# Geophysics-Inspired Nonlinear Stress–Strain Law for Biological Tissues and Its Applications in Compression Optical Coherence Elastography

**DOI:** 10.3390/ma17205023

**Published:** 2024-10-14

**Authors:** Vladimir Y. Zaitsev, Lev A. Matveev, Alexander L. Matveyev, Anton A. Plekhanov, Ekaterina V. Gubarkova, Elena B. Kiseleva, Alexander A. Sovetsky

**Affiliations:** 1A.V. Gaponov-Grekhov Institute of Applied Physics of the Russian Academy of Sciences, Uljanova St., 46, Nizhny Novgorod 603950, Russia; lev@ipfran.ru (L.A.M.); matveyev@ipfran.ru (A.L.M.); alex.sovetsky@ipfran.ru (A.A.S.); 2Privolzhsky Research Medical University, 10/1 Minin and Pozharsky Sq., Nizhny Novgorod 603005, Russia; strike_gor@mail.ru (A.A.P.); kgybarkova@mail.ru (E.V.G.); kiseleva84@gmail.com (E.B.K.)

**Keywords:** optical coherence elastography, nonlinear elasticity, stress–strain law, biomechanics

## Abstract

We propose a nonlinear stress–strain law to describe nonlinear elastic properties of biological tissues using an analogy with the derivation of nonlinear constitutive laws for cracked rocks. The derivation of such a constitutive equation has been stimulated by the recently developed experimental technique—quasistatic Compression Optical Coherence Elastography (C-OCE). C-OCE enables obtaining nonlinear stress–strain dependences relating the applied uniaxial compressive stress and the axial component of the resultant strain in the tissue. To adequately describe nonlinear stress–strain dependences obtained with C-OCE for various tissues, the central idea is that, by analogy with geophysics, nonlinear elastic response of tissues is mostly determined by the histologically confirmed presence of interstitial gaps/pores resembling cracks in rocks. For the latter, the nonlinear elastic response is mostly determined by elastic properties of narrow cracks that are highly compliant and can easily be closed by applied compressing stress. The smaller the aspect ratio of such a gap/crack, the smaller the stress required to close it. Upon reaching sufficiently high compressive stress, almost all such gaps become closed, so that with further increase in the compressive stress, the elastic response of the tissue becomes nearly linear and is determined by the Young’s modulus of the host tissue. The form of such a nonlinear dependence is determined by the distribution of the cracks/gaps over closing pressures; for describing this process, an analogy with geophysics is also used. After presenting the derivation of the proposed nonlinear law, we demonstrate that it enables surprisingly good fitting of experimental stress–strain curves obtained with C-OCE for a broad range of various tissues. Unlike empirical fitting, each of the fitting parameters in the proposed law has a clear physical meaning. The linear and nonlinear elastic parameters extracted using this law have already demonstrated high diagnostic value, e.g., for differentiating various types of cancerous and noncancerous tissues.

## 1. Introduction

The high diagnostic value of mechanical properties of biological tissues has become widely appreciated and confirmed by numerous experimental demonstrations. The experimental techniques for studying tissue biomechanics use various principles and allow for studying a broad range of spatial scales starting from subcellular level to entire macroscopic organs. Some of these methods may be better suited for highly controllable laboratory measurements. For example, Atomic Force Microscopy (AFM) enables a very high subcellular resolution to study the mechanical properties of individual cells and organelles [1,2]. However, AFM is not suitable for studying elastic properties of tissue areas on scales corresponding to groups of cells and, moreover, on larger scales. There is a family of macroscopic methods, for example, indentation-based ones [3], which give information about tissue elasticity on a scale of several millimeters and even larger. Similar and even greater macroscopic scales are characterized in tensile mechanical tests, e.g., [4,5] which usually require rather special preparation procedures (e.g., fabrication of samples with a special “dog-bone” shape). 

There are also known macroscopic elastographic techniques based on ultrasound (US) and magnetic resonance imaging (MRI), which enable spatially resolved characterization of tissue elasticity [6,7]. In contrast to these indentation-based methods and tensile tests, the US-based and MRI-based elastographic techniques are mostly intended for in vivo applications. Presently, US elastography (USE) has become routinely used in clinic applications and has proven its high utility for detection and visualization of tumors, most notably for breast cancer diagnostics in patients [8,9,10].

In view of this, USE methods are probably the most widely known and are presently implemented in various medical ultrasound platforms. Two main approaches utilizing essentially different principles can be pointed out among USE methods. One approach [11,12] comprises wave-based methods in which the shear modulus *G* is estimated by measuring the velocity of fairly slow shear/surface waves. The shear waves are excited using auxiliary focused beams of ultrasound characterized by much greater propagation velocity and the shear-wave propagation is also visualized ultrasonically. This approach, for example, has found widespread application in the diagnosis of chronic liver diseases [13,14]. Another group of USE methods utilizes the so-called compression principle proposed even earlier than the wave-based USE [8]. Compression USE is based on visualization of quasistatic strains produced in the tissue by approximately uniaxial stress that occurs in the vicinity of the ultrasound probe surface acting as a compressing piston. In the framework of the linear elasticity paradigm, the strains produced by such compression should be inversely proportional to the Young modulus *E*. For soft biological tissues with the Poisson’s ratio ν close to the “liquid” limit ν = 0.5, there is a simple relationship: *E* = 3*G* [15]. Thus, for soft biotissues, moduli *E* and *G* give equivalent diagnostic information. However, unlike quantified shear modulus *G* enabled by measuring the shear-wave speed in the wave-based USE, conventional compression USE gives only the relative distribution of modulus *E* in the visualized region, which significantly limits the clinical use of qualitative (not quantitative) compression USE; when assessing the mechanical properties of the lesion, it is necessary to compare with reference regions of normal tissues [16].

In what follows, we focus on another elastographic technique, termed Optical Coherence Elastography (OCE), which emerged during the recent years. Basic structural scans in Optical Coherence Tomography (OCT) [17] resemble US scans but enable significantly higher resolution—several micrometers—although, correspondingly, accompanied by a smaller visualization depth (up to 1–2 mm) and lateral scan size (typically, several mm). Presently OCT has become the “gold standard” in ophthalmic visualization because the resolution of structural OCT fills the gap between high-resolution microscopy and US visualization [18]. For elasticity characterization, OCE similarly fills the niche between high-resolution AFM and macroscopic US elastography in terms of the visualized region size and resolution. 

By analogy with USE, OCE methods also develop in two main directions. One is wave-based OCE in which OCT is applied to visualize auxiliary shear or surface elastic waves, velocities of which are used to estimate shear modulus *G* [19]. Another main direction is compression OCE (C-OCE) in which, similar to compression USE, compression-induced strains are visualized by analyzing OCT scans. However, instead of correlational tracking of displacements widely used in compression USE, C-OCE mostly utilizes phase-resolved strain estimation [20,21]. Furthermore, C-OCE enables quantitative estimation of elasticity (Young’s modulus) due to utilization of precalibrated linearly elastic reference silicone layers placed on the sample surface to play the role of optical stress sensors without the need of additional force sensors [20,22].

C-OCE also demonstrated its ability to directly obtain nonlinear stress–strain dependences for compressed samples [22,23]. Then, by estimating slopes of the stress–strain curves, the tangent Young’s modulus can be estimated for a pre-chosen applied stress level, as described in [24] and demonstrated for nonlinear-elastic tumor tissues [25,26]. Moreover, even the nonlinearity parameter can be estimated for various levels of applied stress and strain in the tissue and used for diagnostics [27,28]. These studies demonstrated that the additional information about tissue nonlinearity may significantly improve differentiation of tissue types/states, which attracts increasing attention in USE as well [29,30,31,32,33]. 

Although C-OCE does not reach cellular resolution typical of histological images, using the revealed differences in the linear and nonlinear elastic properties among various histological structures, C-OCE enables automated morphological segmentation of heterogeneous cancerous tissues with a spatial resolution of several tens of microns. The results of such C-OCE-based segmentation are highly consistent with conventional manual segmentation of histological slides [25,26,27,34]. 

In these C-OCE-based diagnostic procedures, a very important issue is adequate fitting of initial stress–strain curves reconstructed using C-OCE. To perform fitting of experimental nonlinear dependences, various nonlinear laws are known from the literature and are widely used to interpret elastic behavior of various tissues [5]. The form of such empirically proposed models is often supported by symmetry/invariance considerations, so that such models are fairly universal, which is simultaneously their strength and weakness. Indeed, usually these models do not directly reflect microstructural features of tissues, although this microstructure often is of primary interest for diagnostic conclusions. This explains why attempts are made to construct models relating elastographic data with tissue structure [35,36]. However, creation of models based on detailed considerations of microstructure usually limits their applicability to a rather narrow class of tissues. 

Actually, a similar problem of reasonable balance between universality of empirical models and narrower applicability of particular microstructure-based models arises in other areas of physics. For example, in rock physics, it has been broadly accepted for decades (e.g., [37,38,39,40]) that the nonlinear elasticity of rocks is essentially determined by their microstructure, first of all, the presence of narrow, highly compliant cracks in the host material. Such cracks cause a pronounced decrease in the elastic modulus of rocks. At the same time, narrow cracks also lead to pronounced elastic nonlinearity of the material because applied compression gradually closes cracks starting from the most narrow and simultaneously most compliant ones. As a result, the elastic modulus may pronouncedly increase with increasing compressive stress and tend toward the value typical for the homogeneous host material. It is important that, for cracked materials, this pronounced increase in the modulus value occurs for quite small strains, often on a scale of several percent and even smaller. Such small strains may seem negligibly small in comparison with elongations of soft biological tissues in tensile tests, which are often used to understand nonlinear biomechanical properties. In the models used for interpretation of tensile tests, the deformation is usually characterized in terms of tissue elongation with respect to its initial state and, for soft biotissues studied in such experiments, the elongations often reach tens of percent and greater (e.g., [5,41]). Correspondingly, at such large deformations, even purely geometrical factors (unrelated to tissue microstructure) give an important contribution to the observed tissue nonlinearity (see, e.g., classical book of Fung [42]). 

At the same time, results of independent indentation/compression experiments related to studying nonlinear behavior of biotissues (e.g., [3,22,23,27]) indicate that many tissues demonstrate a pronounced (up to several times and greater) increase in the elastic modulus for quite moderate compressive strains, 1–10%, whereas soft polymers often used as biotissue phantoms do not exhibit similar pronounced nonlinearity in this strain range. The abovementioned results for biotissues qualitatively are very similar to the compression dependence of elastic moduli for cracked rocks. This fact suggests that such similarity of nonlinear-elastic behaviors of rocks and biotissues may be caused by similar structural features.

In what follows, we first recall the principles of quantitative C-OCE. Up until now, this technique has been discussed exclusively in publications related to the development of biophotonics techniques and their applications to biomedical diagnostics, for which C-OCE-based assessment of nonlinear elasticity has proven its very high usefulness. Then, we present experimentally supported arguments that, using analogies with rock physics, the nonlinear elastic properties of biotissues observed by C-OCE in compression tests can be explained by compression-produced closing of narrow interstitial gaps/pores. Using this idea, we propose a formulation of nonlinear stress–strain law by analogy with models of cracked rocks and demonstrate that the derived stress–strain relationship enables very good fitting of data for quite diverse tissues. Furthermore, we demonstrate that the proposed stress–strain law allows for obtaining some quantified conclusions about tissue microstructure that is not directly resolved in OCT images, but in certain cases reasonably agrees with the high-resolution histological images. The proposed stress–strain law has already proven to be very useful for fitting experimental data. The subsequent analysis of the fitted stress–strain curves allows for formulation of efficient diagnostic criteria in terms of linear and nonlinear elastic parameters of biotissues, which in turn enables automated segmentation of C-OCE scans that demonstrates a striking similarity with conventional segmentation of histological images. 

Thus, in Section 2.1 that follows, we first briefly recall the principles of obtaining nonlinear stress–strain curves in the recently developed phase-sensitive compression OCE. Then, in Section 2.2, we present as background some experimentally supported arguments, in which we derive a nonlinear stress–strain law for biotissues by analogy with models of cracked rocks in geophysics. Finally, in Section 3, the efficiency of the proposed law for fitting C-OCE-based experimental data obtained for a broad range of various biological tissues is demonstrated. 

## 2. Materials and Methods

### 2.1. Acquiring Experimental Nonlinear Stress–Strain Curves Using C-OCE

Unlike correlational displacement tracking widely used in USE for subsequent strain estimation, in C-OCE, phase-resolved analysis of initial complex-valued structural OCT scans is mostly used. Details of formation of OCT signals based on principles of low-coherence interferometry can be found in review [43]. For the present consideration, it is sufficient to point out that each pixel in the structural OCT scan is characterized by its amplitude and phase. The typical distance between neighboring pixel centers in the axial and lateral direction is several micrometers. Schematically, the typical configuration of an OCT probe contacting with the studied tissue through the reference layer of translucent silicone required for elasticity quantification is shown in Figure 1a. The other images correspond to an actual breast cancer sample similar to those characterized by C-OCE in studies [25,27] with histological confirmation. An example of a typical 2D OCT scan is shown in Figure 1b (as an intensity image in dB scale). Figure 1c shows the color-coded phase difference between the subsequently acquired OCT scans of the tissue compressed by the output glass of the OCT probe.

Axial displacements U of scattering particles in a medium with fairly uniform distribution of the refractive index n are connected with the interscan phase variations Φ by a well-known relationship: (1)$$U=\frac{\lambda_{0}\Phi }{4\pi n}$$
where $\lambda_{0}$ is the central optical wavelength of the OCT source in vacuum. Therefore, the interscan axial strain s=dU/dz is readily found as the axial gradient of the interscan phase variation: (2)$$s=\frac{\lambda_{0}}{4\pi n}\frac{\partial \Phi }{\partial z}$$

The axial strain distribution corresponding to the interscan phase difference in Figure 1c is shown in Figure 1d. In practice, the numerical differentiation in Equation (2) can be performed, e.g., using the least-square estimation of Φ(z)-slope [44], which requires phase unwrapping for wrapped phase-variation maps similar to the one shown in Figure 1c. 

Alternatively to the least-square fitting, the so-called “vector” approach can be applied, which does not need phase unwrapping [45,46]. The name “vector” reflects the fact that this method operates with complex-valued quantities as vectors in the complex plane without singling out the phase until the very last stage. An important advantage of this method is that it enables estimation of local phase-variation gradients, for which the wrapping-related ambiguity of total phase variation is not important. 

It should be noted that the applicability of Equation (2) based on pixel-to-pixel comparison of two OCT scans is limited to interframe strain below ~1–2% (since larger strains cause excessive decorrelation of the compared scans). Nevertheless, the phase approach enables estimation of much larger cumulative strains S up to several tens of percent. Such strains can be evaluated via summation of incremental interframe strains found via Equation (2) [47]: (3)$$S=\sum_{i}s_{i}$$

When the strain-induced displacements reach suprapixel magnitudes, tracking of displaced particles may be required for correct estimation of local stiffness [48]. In such a way, quantitative strain can be correctly mapped in both homogeneous reference silicone and mechanically inhomogeneous tissue. 

The next key step in the described C-OCE technique is acquiring stress–strain curves using the reconstructed interscan strain distributions and their summation. In this context, it is of key importance to emphasize that the reference silicone is elastically highly linear. More specifically, the term “linear” means that if the silicone layer is strained by a small incremental strain ds, the developed incremental stress dσ is given by $d\sigma =E_{sil}ds$, where Young’s modulus $E_{sil}$ of the silicone remains invariable ($E_{sil}=const.$) independent of the current cumulative strain of the silicone layer. This linearity can readily be verified experimentally by compressing a sandwich of silicones with strongly different Young’s moduli. The sandwich layers experience the same applied stress, but the resulting cumulative strain given by Equation (3) is very different for these layers. If the layers are elastically linear, i.e., their Young’s moduli are invariable, the cumulative strain in one layer and another layer with a strongly different modulus should remain linearly proportional. The performed experiments confirmed this expectation for such silicone layers with contrasting stiffness. With good accuracy, the two cumulative strains exhibited linear proportionality up to S~50–70% in one layer, whereas in the other layer the strain was much smaller, for example, S~5–10% [22,47].

Consequently, the current stress in silicone (applied to the underlying tissue) can be readily estimated via cumulative strain in the reference silicone:(4)$$\sigma =\sum_{i}d\sigma_{i}=E_{sil}\sum_{i}s_{i}=E_{sil}\cdot S$$

Then, plotting cumulative strain in the pre-calibrated silicone recalculated to stress via Equation (4) against cumulative strain in the underlying tissue, one obtains the stress–strain relationship for the tissue. Examples of the stress–strain curves determined for an actual breast cancer sample are shown in Figure 1e as the dashed curves. The latter correspond to regions marked by labels 1 and 2 in the strain map shown in Figure 1d. Although formally such stress–strain curves can be plotted for every pixel in the tissue, for improving the signal-to-noise ratio, the strains are estimated with averaging over rectangular areas of ~80–100 μm in size, which determines the resolution in elastographic maps similar to Figure 1d.

It is clear that the stress–strain curves in Figure 1e exhibit pronounced nonlinearity, which is a rule for most biotissues rather than an exception. The tangent Young’s modulus $E_{tg}=d\sigma /ds$ for the tissue, i.e., the slope of the stress–strain curve is, therefore, pronouncedly dependent on current stress and strain in the tissue. The solid lines in Figure 1e show the fitting curves, which can be readily differentiated to evaluate the tangent Young’s modulus $E_{tg}$ as a function of the current stress and corresponding strain in the tissue. The Young’s modulus also demonstrates pronounced dependence on stress and strain. The next derivative $\beta =dE_{tg}/d\sigma$ is dimensionless and has the meaning of a local quadratic nonlinear parameter corresponding to the current stress (or strain). These curves for the nonlinearity parameter β are shown in Figure 1g. Finally, Figure 1h–j show the spatially resolved maps for the tangent modulus $E_{tg}$ in the tissue estimated for three “standardized” stress levels: 1 kPa, 5 kPa and 10 kPa. 

We point out that initial strain maps (as shown in Figure 1d) are characterized by pronounced spatial inhomogeneity of stress within the visualized area. This stress inhomogeneity is caused by combined influence of nonideally uniform thickness of the silicone, nonideally planar surface of the tissue and inhomogeneity of its mechanical properties. This inhomogeneity causes uncontrollable variability of the tangent modulus for the examined nonlinear tissues, so that for meaningful and reproducible comparison of the tangent Young’s modulus $E_{tg}=E(\sigma )$, the latter should be estimated for a “standardized” stress σ using a series of OCT scans of the compressed tissue (see details in [24]). These maps of $E_{tg}$ in Figure 1h–j clearly demonstrate that the tissue nonlinearity causes strong variations in the Young’s modulus values and strongly affects the geometrical shape of the stiffest region corresponding to the tumorous tissue. Consequently, to differentiate morphological components of the tissue, the threshold values $E_{th}(\sigma )$ of the Young’s modulus should be determined for a specific “standardized” stress level using comparison of C-OCE maps with the corresponding histological images [25,27,28]. After determining the characteristic ranges of $E_{th}$, the C-OCE images can be automatically segmented, demonstrating a very high correlation with manually performed morphological segmentation of histological images [26]. However, it was found that in some cases the use of tangent Young’s moduli may be insufficient to reliably differentiate various tissue components. In such cases, the simultaneous usage of the Young’s modulus and nonlinearity parameter can be helpful (see details in refs. [27,28]).

The description of C-OCE principles and examples of their application to real biotissues presented above clearly demonstrate that utilization of the linear-elasticity paradigm is strongly insufficient. Even for apparently small compressive strains, about a few percent, the tangent (current) Young’s modulus may vary several times. This is not typical of tensile tests, where comparably strong manifestations of nonlinearity are often observed for elongations of tens and even hundreds of percent. The examples presented above also demonstrate that estimations of slopes of experimental dependences are very important, so that utilization of adequate fitting laws can be very helpful. In the next section, we describe such a dependence, the formulation of which was initially inspired by the C-OCE results obtained for a special case of tissue containing narrow, crack-like pores, but afterward the derived stress–strain law demonstrated a surprisingly good quality of fitting for a rather broad range of various biotissues.

### 2.2. Derivation of Stress–Strain Law Describing Elastic Nonlinearity of Tissues Containing Highly Compliant Gaps/Pores

Despite quite a broad range of earlier proposed constitutive laws for biotissues, e.g., those used in [5,23], it was not easy to choose a variant, in which several features typical of nonlinear responses observed with C-OCE would be clearly caught, allowing for a physically meaningful interpretation of the fitting parameters. In particular, this relates to C-OCE examinations of such samples as corneal tissues characterized by the clear presence of collagenous layers with a rather high Young’s modulus in the MPa range typical of collagen fibrils and bundles (which is known from AFM results, e.g., [49,50,51]). At the same time, corneal tissue examined using C-OCE demonstrated the macroscopic modulus in the order of tens of kPa at the very beginning of compression and rapid increase toward MPa range for quite moderate compressive strains, ~5–7% [48]. These observations suggested that narrow gaps/pores occurring between stiff collagen layers could be closed by the applied compressive stress, leading to a strong increase in the elastic modulus. 

Geophysicists long ago realized that even a fairly small volume content of narrow pores/cracks may strongly reduce the elastic modulus of rocks, and may cause its pronounced dependence on the applied stress. Indeed, a crack with diameter D and opening h with a small aspect ratio
(5)α~h/D<<1
occupies the actual volume ~$D^{2}h$; however, such a crack releases the elastic energy of a strained material in the much larger volume ~$D^{3}$. Thus, their influence on the material modulus may be very pronounced, comparable with the influence of spheroidal pores of the same diameter D. Each of these pores occupies volume $\sim D^{3}$ that is much larger than the volume of narrow cracks with the same D. 

For the uniaxial stress considered typical for C-OCE examinations, it is important that such narrow pores are easily closed by the normal component of the applied compressive stress
(6)$$\sigma_{clos}\sim \alpha E_{m}$$
where $E_{m}$ is the elastic modulus of the matrix material [38]. Consequently, to completely close such a pore, it is sufficient to create in the material an average strain
(7)$$s_{clos}\sim \alpha <<1$$

Here, possible factors near the order of unity are omitted, so that independent of the fine geometric details of thin cracks, this statement can be considered as a rule of thumb [52]. For cracks in rigid rocks, aspect ratios are very small (often < $10^{-3}{\ldots 10}^{-4}$, [53]). Although the criteria for crack-closing strains and stress were reliably formulated theoretically long ago [38,52], to the best of our knowledge, there has not been direct visualization of crack closure during material straining. 

In this context, it is interesting that the OCT visualization described in the previous section opens the possibility to directly visualize stress-produced closing of crack-like delaminations in soft biotissues. Such an example is demonstrated in Figure 2 for a pericardium sample (pericardium samples are discussed in more detail in Section 3.2). The sample discussed here was dried and then again impregnated by saline solution. The drying evidently induced crack-like delaminations between pericardium layers visible in the OCT images. One of these crack-like defects is shown in Figure 2. It is worth noting that during the applied quasistatic loading, the saturated liquid was easily squeezed out from the crack to the surrounding tissue and did not impede closing of the crack as if it were a dry crack. 

In the initial state (Figure 2a), the crack-like defect is open with aspect ratio α~0.1. During compression loading, the defect demonstrated gradual closing down to virtually complete closure, as shown in Figure 2b. By reaching this state, the closing strain $s_{clos}$ estimated in the host material within the rectangle area indicated in Figure 2a,b corresponded to the value $s_{clos}\sim \alpha \sim 0.1$, in agreement with Equation (7). 

Comparison with strain in the pre-calibrated reference silicone indicated that the Young’s modulus $E_{m}$ in the matrix (host) pericardium was $E_{m}\sim 200$ kPa, whereas the crack became closed already for applied stress $\sigma_{clos}\sim 20$ kPa, corresponding to the strain in the matrix tissue $s_{clos}\sim 0.1$ (which agrees with Equations (6) and (7), bearing in mind that α~0.1). Indeed, the stress–strain curve in Figure 2d is obtained outside the crack-like defect for the sample region marked by the rectangle in Figure 1a–c, where the material is fairly homogeneous and linear. The measured tissue strain in this region for the moment of closing is $s_{clos}\sim 0.1$. For the crack with initial aspect ratio α~0.1, this strain agrees the theoretical expectation (Equation (7)). At the same time, the slope of the stress–strain curve in Figure 2d indicates that the Young’s modulus of the matrix is $E_{m}\sim 200$ kPa, whereas much smaller stress $\sigma_{clos}\sim 20$ kPa causes the crack to close, which is in agreement with Equation (6).

In other words, one can say that narrow crack-like gaps with α<<1 act as high compliant (soft) inclusions that experience 100% straining when the compressed host material experiences only fairly small strain s~α<<1. In view of this, such cracks can also be considered as effective soft inclusions with a strongly reduced effective elastic modulus $E_{incl}=\varsigma E_{m}<<E_{m}$, for which comparison with Equations (1) and (3) indicates that the compliance parameter ς for such inclusions is
(8)$$\varsigma \sim \alpha \sim s_{clos}$$

Considering an elastic matrix with modulus $E_{m}$ containing arbitrary (but identical) soft inclusions characterized by reduced modulus $E_{incl}=\varsigma E_{m}$ with the total volumetric content $\upsilon_{t}$ per unit volume, very simple considerations of accumulated elastic energy yield the following expression for the effective reduced modulus $E_{eff}$ for such a heterogeneous material [54]:(9)$$\frac{E_{m}}{E_{eff}}=1+\frac{\upsilon_{t}}{\varsigma }$$

For example, if the role of such soft inclusions is played by conventionally discussed narrow cracks, then according to Equations (5) and (6) the compliance parameter is ς~h/D, whereas the volume of one crack is approximately $hD^{2}$. Thus, for crack concentration $n_{cr}$, their volumetric content per unit volume in Equation (9) is $\upsilon_{t}\sim n_{cr}hD^{2}$. Equation (9) then yields
(10)$$\frac{E_{m}}{E_{eff}}\sim 1+n_{cr}D^{3}$$

With an accuracy by a factor near the order of unity before $n_{cr}D^{3}$, this expression agrees with the results of many authors for elasticity of cracked materials (e.g., [37,55,56]). Such accuracy is sufficient for the following consideration. 

Certainly, in real material (either rocks or biological tissues), narrow pores/cracks are not identical in their parameters. They are characterized by a distribution υ(ς) over their compliance parameters and aspect ratios α~ς and, correspondingly, over closure stresses $\sigma_{clos}$, which is related to α by Equation (6). Bearing this in mind, in the case of inclusions characterized by a distribution υ(ς) over their compliance parameters instead of Equation (9), one can write
(11)$$\frac{E_{m}}{E_{eff}}=1+\int_{\varsigma_{\min}}^{\varsigma_{\max}}\frac{\upsilon (\varsigma )}{\varsigma }d\varsigma$$

Here, function υ(ς) is dimensionless with normalization to the total volumetric content per unit volume $\upsilon_{t}$ of all crack-like defects:(12)$$\int \upsilon (\varsigma )d\varsigma =\upsilon_{t}$$

In terms of $\upsilon_{t}$ given by Equation (12) for nonidentical inclusions and the corresponding average (characteristic) value of the aspect ratio $\alpha_{av}$ and compliance parameter $\varsigma_{av}$ of the crack-like defects (with $\alpha_{av}\sim \varsigma_{av}$ according to Equation (8)), Equation (11) can be rewritten in the following form similar to Equation (9) for identical defects:(13)$$\frac{E_{m}}{E_{eff}}=1+\frac{\upsilon_{t}}{\varsigma_{av}}$$

In Equation (13), according to Equations (11) and (12), the average compliance parameter $\varsigma_{av}$ is defined as $1/\varsigma_{av}=\upsilon_{t}^{-1}\cdot \int_{\varsigma_{\min}}^{\varsigma_{\max}}[\upsilon (\varsigma )/\varsigma ]d\varsigma$. 

The utility of Equation (9) and its counterpart (13) for real nonidentical defects was demonstrated in study [57] for estimating the characteristic (average) aspect ratio $\alpha_{av}\sim \varsigma_{av}$ of narrow crack-like pores (gaps) arising among collagen layers in samples of collagenous tissues—rabbit cornea and costal porcine cartilage. These samples were subjected to moderate pulse-periodic heating (up to 55–65 °C) by infrared laser irradiation, which caused breaking of some intermolecular links and formation of narrow interstitial gaps that could be histologically visualized. Such laser heating is used in emerging biologically nondestructive methods for reshaping cartilage samples used in the fabrication of cartilaginous implants and corneal refraction correction [58,59,60]. The visualization of thermal strains caused by laser heating in study [57] was monitored using the OCE method described in Section 2.1. The layers of collagen in the corneal and cartilaginous samples were oriented orthogonally to the OCT beam. Thus, the OCE monitoring made it possible to estimate the heating-induced axial cumulative strain caused by the formation of interlayer gaps. In other words, it was possible to evaluate specific content $\upsilon_{t}$ of post-heating laser-induced narrow pores (interstitial gaps) that were induced among the collagenous layers. This estimation of $\upsilon_{t}$ was then combined with C-OCE estimation of the complementary post-heating modulus reduction in the irradiated area in comparison with the modulus in the surrounding nonirradiated regions. In these modulus estimations, delicate compression was applied to the samples through the reference silicone layer. The compression-induced strains in the tissue were <1%. Thus, the ratio $E_{m}/E_{eff}$ of the moduli was estimated for nonirradiated ($E_{m}$) and irradiated areas ($E_{eff}$). The ratio $E_{m}/E_{eff}$ corresponding to the modulus reduction induced by the appearance of the soft crack-like pores was quite significant, up to 2–3 times for fairly small $\upsilon_{t}$, in the order of several percent [57]. The two independently estimated parameters $\upsilon_{t}$ and $E_{m}/E_{eff}$ could be substituted in Equation (13) to estimate the average compliance parameter and aspect ratio of the laser-induced narrow pores $\alpha_{av}\sim \varsigma_{av}$. Thus, it was found that $\alpha_{av}\sim \varsigma_{av}\sim 1/30\ldots 1/20$. The structural OCT images did not allow for direct verification of these conclusions because the laser-induced pores were not yet resolved, unlike the fairly large crack-like pore shown in Figure 2. However, the OCE-based conclusions in study [57] were corroborated by high-resolution histological images of corneal samples The histology directly demonstrated the appearance of such crack-like pores in irradiated samples and their absence in nonirradiated samples.

Now, after demonstrating utility of the idea about the role of narrow pores in biotissues in the elastic-modulus reduction observed in the small strain range (i.e., a fairly linear regime for the matrix material), one may derive the expected form of nonlinear stress–strain dependences. Recall that narrow cracks should be gradually closed by increasing compression strain. It is clear that such increasing tissue compression initially affects the most compliant narrowest pores. In other words, with increasing compression the lower limit $\varsigma_{\min}$ in Integral (11) shifts toward larger values. In contrast, the stiffest pores are not yet affected. Therefore, the upper limit in Integral (11) remains unaffected (and formally can be set to infinity). Recalling that the compliance parameter and the aspect ratio of narrow pores are approximately equal, α~ς, Integral (11) can also be represented via pore distribution υ(α) over their aspect ratios (also with normalization $\int \upsilon (\alpha )d\alpha =\upsilon_{t}$):(14)$$\frac{E_{m}}{E_{eff}}=1+\int_{\alpha_{\min}}^{\infty }\frac{\upsilon (\alpha )}{\alpha }d\alpha$$

Now, we point out that in Equation (14), the lower limit $\alpha_{\min}$ corresponds to the current stress σ, for which all compliant defects with $\alpha <\alpha_{\min}$ are already closed, as schematically illustrated in Figure 3.

For direct experimental comparison, it is more convenient to reformulate Integrals (11) and (14) via the distribution υ(σ) of the compliant inclusions over the closing stress:(15)$$\frac{E_{m}}{E_{eff}}=1+E_{m}\int_{\sigma }^{\infty }\frac{\upsilon (\sigma \prime )}{\sigma \prime }d\sigma \prime$$

To obtain Equation (15) from Equation (4), we accounted for Equation (6), which relates the closing stress with aspect ratio. This explains the appearance of an additional dimensional factor $E_{m}$ before the integral in Equation (15). Indeed, unlike dimensionless distributions υ(ς) and υ(α), the distribution υ(σ) over the closing stress has the dimensionality of inverse stress to enable correct normalization condition $\int \upsilon (\sigma \prime )d\sigma \prime =\upsilon_{t}$. Thus, due to factor $E_{m}$ before the integral in Equation (15), both sides of this equation have correct dimensionless form. 

Equation (15) indicates that with increasing stress σ, the current (tangent) elastic modulus $E_{eff}$ increases and tends toward value $E_{m}$ of the matrix material because of gradual closing of the narrow pores, as illustrated in Figure 3. It is clear from Equation (15) that if the stress dependence of the Young’s modulus $E_{eff}(\sigma )$ is experimentally determined, then differentiation of Equation (15) with respect to σ allows for reconstructing distribution υ(σ):(16)$$\frac{d}{d\sigma }\left(\frac{1}{E_{eff}}\right)=-\upsilon (\sigma )/\sigma$$

According to Equation (6), bearing in mind that the closing stress $\sigma_{clos}$ is determined by the aspect ratio of compliant defects, and the distribution υ(σ) obtained from Equation (16) is proportional to distribution υ(α) of the defects over their aspect ratios. Similar procedures in using experimentally measured stress dependences of elastic moduli were proposed in rock physics for obtaining crack distributions over their aspect ratios, e.g., [38]. 

In principle, relationship (16) does not require any a priori assumptions about the reconstructed distribution υ(σ). However, numerical differentiation of experimentally found dependences in actual noisy conditions is rather error prone, so that it would be advantageous to use procedures that are more robust to measurement errors. For the utilization of C-OCE technique described in Section 2, such procedures can be based on proper fitting of the experimentally reconstructed stress–strain curves σ(s). 

In view of the abovementioned analogy with rock physics, one may recall that pressure dependences of elastic moduli for rocks usually can be well fitted by exponential functions, e.g., [39]. It is clear from the structure of Equation (15) that the right-hand side of Equation (15) exhibits exponential behavior if the distribution υ(σ) has the following bell-shape form:(17)$$\upsilon (\sigma )=(\upsilon_{t}/B^{2})\sigma \exp(-\sigma /B)$$
where, as before, $\upsilon_{t}=\int_{0}^{\infty }\upsilon (\sigma )d\sigma$ and maximum of υ(σ) corresponds to the characteristic stress σ=B. 

Next, we recall that the tangent modulus corresponds to the slope of stress–strain dependence $E_{eff}=d\sigma /ds$. Then, combining the latter relation with Equations (15)–(17), one obtains
(18)$$E_{m}ds=[1+E_{m}\frac{\upsilon_{t}}{B}\exp(-\sigma /B)]d\sigma$$

Bearing in mind that s=0 for σ=0, integration of Equation (18) yields the following stress–strain relationship, written as s=s(σ):(19)$$s=\frac{\sigma }{E_{m}}+\upsilon_{t}[1-\exp(-\sigma /B)]$$

Again, recalling that $E_{eff}(\sigma )=d\sigma /ds$ and performing differentiation of Equation (19), one readily obtains the following expression for the tangent Young’s modulus $E_{eff}$:(20)$$E_{eff}(\sigma )=\frac{E_{m}}{1+(\upsilon_{t}E_{m}/B)\exp(-\sigma /B)}$$

Equation (20) means that for increasing stress, which gradually closes highly compliant crack-like pores/gaps, the tissue modulus tends toward the modulus $E_{m}$ of the matrix material. For small pressures, modulus $E_{eff}(\sigma )$ may be significantly reduced compared to the matrix value $E_{m}$. This strongly resembles the elasticity behavior of cracked rocks [38]. Next, one can point out that derivative $\beta =dE_{eff}(\sigma )/d\sigma$ is dimensionless and in the vicinity of any current stress σ it takes the form
(21)$$\beta \equiv \frac{dE_{eff}(\sigma )}{d\sigma }=\frac{E_{m}}{B}\frac{\left(\upsilon_{t}E_{m}/B)\exp(-\sigma /B\right)}{{\left[1+(\upsilon_{t}E_{m}/B)\exp(-\sigma /B)\right]}^{2}}$$

Expression (21) has the meaning of a dimensionless parameter of quadratic nonlinearity (since β determines local parabolic approximation of the nonlinear stress–strain dependence (19) in the vicinity of stress σ). According to Equation (21), the nonlinearity parameter β(σ) may either monotonically decrease with increasing stress σ or may have an intermediate maximum. Indeed, denoting $(\upsilon_{t}E_{m}/B)\exp(-\sigma /B)=y$, Equation (13) takes the form
(22)$$\beta =\frac{E_{m}}{B}\frac{y}{{\left[1+y\right]}^{2}}$$
with the maximum $\beta =(E_{m}/4B)$ corresponding to y=1. It is clear that quantity y decreases with increasing σ, so that β(σ) may have an intermediate maximum at a non-zero stress if for zero stress σ=0
(23)$$\upsilon_{t}E_{m}/B>1$$

Typically, the total content of narrow pores $\upsilon_{t}<<1$, the condition in (23) requires that the ratio $\left(E_{m}/B\right)$ is sufficiently large. Recalling Equation (20), we conclude that the existence of intermediate maximum of nonlinearity parameter β(σ) requires that $E_{eff}(\sigma =0)<E_{m}/2$. In other words, initially for σ=0, the presence of narrow pores reduces the tissue modulus more than twice in comparison with the matrix modulus. 

We recall that the bell-shape distribution given by Equation (17) is written through an analogy with rock physics and is not supported by arguments specifically related to biological tissues. Nevertheless, it is demonstrated in Section 3 that Equation (19), based on the assumed Equation (17), enables surprisingly good fitting of experimental stress–strain relationships obtained using the C-OCE technique for compressive loading of rather diverse biological tissues. 

## 3. Results of Applying the Proposed Stress–Strain Law for Description of Nonlinear Response of Various Biotissues

In this section, we present several examples demonstrating the usefulness of the discussed analogy with rock physics and the assumption that for biological tissues their nonlinear stress–stress behavior under compressive loading is caused by gradual closing of soft porosity initially existing in the tissue. In this regard, our basic assumptions are also strongly supported by results of study [61], in which high-resolution histological examinations revealed unrecognized interstitium in various human tissues (corresponding to the system of pores/gaps in the argumentation presented in the previous sections). In the following sections, we present results for seven rather diverse tissue types with 3–7 subtypes and/or states in each category. Where possible, along with C-OCE-enabled stress–strain curves and results of their fitting by the proposed stress–strain law, results of histological examinations are also presented.

### 3.1. Nonlinearity of Eye Cornea under Uniaxial Compression

The first example relates to rabbit’s corneal tissue that was characterized in OCE-based studies [57,62,63]. These studies were stimulated by the discussed in the literature prospects of nonsurgical correction of corneal shape using moderate (up to 55–60 °C) pulse-periodic heating by an infrared laser [64]. It were just the results of cornea-related studies [47,57,62], which initially stimulated us to search an appropriate stress–strain law for interpretation of stress–strain curves for corneal tissue. These curves were obtained for laser-irradiated corneal samples reconstructed using the C-OCE principles described in Section 2.1.

Recall that corneal tissue consists of a stack of nearly plane-parallel sheets of collagenous fibers. In the discussed experiments, these collagen layers were oriented orthogonally to the optical beam of the heating laser and the probing OCT beam during C-OCE examinations. Figure 4a shows schematically the experimental configuration typically used in these studies, and a representative structural OCT scan of a corneal region subjected to pulse-periodic laser heating; details are provided in [57,62,63].

Even with an unaided eye, an area with increased thickness is visible in the heated-zone center shown in Figure 4a. Figure 4c shows two histological images (reproduced from [62]) of a nonheated cornea sample and a sample subjected to heating in a similar regime. For the heated sample, one can clearly see in the histological image fairly large elongated crack-like pores/delaminations parallel to the direction of collagenous layers, whereas in the nonheated sample such pores are not visible. The appearance of such stable pores explains why the heated zone remains dilated after heating. 

Next, by analogy with cracked rocks, it could be expected that the presence of such crack-like pores in the heated region should reduce the Young’s modulus of the tissue. This expectation was confirmed in [57], where this modulus reduction was clearly observed using the C-OCE principle. Furthermore, by applying gradually increasing compressive stress through the layer of pre-calibrated silicone to the post-heating sample, it was possible to obtain spatially resolved stress–strain curves. By analyzing the fitting curves defined by Equation (19), the tangent Young’s modulus could be estimated for a desired stress level. The spatial map of the tangent Young’s modulus (stiffness) in Figure 4b is plotted for 4 kPa stress. The initially reconstructed stress–strain dependences together with the fitting curves for the representative zones 1, 2 and 3 are shown in Figure 4(d-1).

Zone 1 is located fairly far from the heated zone 3 and had almost unaffected properties, whereas in the central heated region 3 the tissue exhibited clear post-heating dilatation. Zone 2 is located at the periphery of the heating zone. Estimation of cumulative strain during the entire heating procedure and subsequent cooling indicated that, after heating, in contrast to the expanded region 3, region 2 clearly experienced shrinkage. The latter was evidently caused by compression in this region, produced by the thermally dilated tissue in the central zone 3. Figure 4b clearly demonstrates that, as a result of these processes, the post-heating stiffness distribution in the initially fairly spatially uniform cornea becomes inhomogeneous. Namely, the dilated corneal tissue in the heated zone 3 becomes pronouncedly softer than in the nonheated zone 1, whereas in the shrinkage zone 2, on the contrary, the tissue becomes stiffer. 

The individual stress–strain curves obtained for zones 1, 2 and 3 are shown in Figure 4(d-1) and in agreement with the stiffness map shown in Figure 4b, which demonstrates that the slope for curve 1 (tangent Young’s modulus for the nonheated zone) is noticeably greater than for the central heated zone 3, but smaller than for the shrunken zone 2. All three curves are very well fitted by Equation (19). Curve 2 for the shrunken zone in Figure 4(d-1) has a narrow nonlinear region for strains <1%, in which the slope of the stress–strain curve rapidly increases, and then tends toward a near-constant value. Curves 1 and 3 also are initially nonlinear (with lower slopes than for curve 2), but with increasing strain their slopes also become nearly invariable and close to the asymptotic slope value for curve 2. This behavior is naturally explained by gradual closing of interstitial soft (micro)pores with increasing applied stress. More specifically, in Figure 4(d-1) for zone 1 in the nonheated tissue, the fitting parameters in stress–strain Equation (19) are: the matrix-tissue stiffness $E_{m}=736$ kPa; the characteristic closing stress B=4.1 kPa; and the initial specific volume content of soft features $\upsilon_{t}=0.026$. For intermediate zone 2, which experienced moderate heating accompanied by compression from the dilated heated zone 3, the matrix-tissue stiffness moderately increased, $E_{m}=839$ kPa; whereas the characteristic closing stress B=0.66 kPa decreased ~7 times and simultaneously the volume content of soft features decreased ~3.8 times, $\upsilon_{t}=0.0068$. Finally, the heated zone 3 demonstrated a slight increase in the matrix-tissue stiffness, $E_{m}=747$ kPa; the characteristic closing stress moderately increased ~1.8 times, B=7.36 kPa, whereas the soft-feature content $\upsilon_{t}=0.069$ pronouncedly increased (~2.7 times) in comparison with nonheated zone 1.

These moderate variations in the asymptotic values $E_{m}$ are clearly seen in Figure 4(d-2) for the Young’s modulus dependence on strain and stress. The magnitudes of parameters B and $\upsilon_{t}$ can also be readily understood from the soft-pore distributions υ(σ) over closing stresses, as shown in Figure 4(e-1). We recall that the total content $\upsilon_{t}$ corresponds to the area under the curve υ(σ), whereas the maximum of υ(σ) occurs exactly for σ=B according to Equation (17). In particular, Figure 4(d-1), based on the analysis of the stress–strain curves, clearly shows that in the heated zone, the strong broadening toward higher closing stresses and a general increase in magnitude for distribution υ(σ) suggests that in zone 1 one should expect the appearance of heating-induced pores with larger aspect ratios (larger opening) according to Equations (6) and (7).

For the discussed case of heated corneal tissue, this expectation for the development of additional interstitial pores with larger aspect ratios (and possibly larger sizes) was possible to confirm histologically. These larger interstitial pores may be visible in histology (see the lower image in Figure 4c). However, even if such pores are not resolved (as in the upper histological image of nonirradiated cornea in Figure 4c), they clearly manifest themselves via the nonlinear elastic response of the tissue. 

The other plots in Figure 4 are based on the analysis of the fitting curves from Figure 4(d-1). In addition to Figure 4(d-2),(d-3) for the dependences of the tangent Young’s moduli as functions of strain and stress, respectively, and Figure 4(e-1) for the reconstructed distribution υ(σ) of pores defined by Equation (17), Figure 4(e-2) shows the nonlinearity parameter β(σ) as a function of stress defined by Equation (21). Figure 4(e-3) shows a parametrical plot demonstrating the complementary evolution of parameters β(σ) and $E_{eff}(\sigma )$ on the plane ($E_{eff,}\beta$).

For the corneal tissue composed of collagenous layers, the existence of interlayer pores/gaps was quite expected, so that the usefulness of the geophysics-inspired stress–strain Equation (19) for fitting experimental stress–strain was not surprising. Less expected was the ability of Equation (19) to enable good-quality fitting for a broad variety of other tissues. In the following sections, we demonstrate that stress–strain curves for six other essentially different tissue types can also be well described by the proposed geophysics-inspired equation of state. Totally, we demonstrate 30 examples of the initially measured stress–strain curves for very diverse tissues. In view of the limited article length, in the other 29 cases we do not provide a detailed discussion of the fitting parameters $E_{m}$, B and $\upsilon_{t}$ as in this section for cornea. However, as pointed out above, their characteristic values and inter-relations can readily be seen from the plots similar to those in Figure 4 for corneal tissue.

### 3.2. Nonlinearity of Pericardium Tissue

The next example relates to another tissue type, namely, human pericardium studied in [65]. This mostly collagenous elastic shell surrounds the hearts of mammalian animals. It has a fairly small thickness (for humans, <1 mm and smaller). The interest in studying the biomechanics of pericardium is stimulated by the fact that it is rather widely used in cardiosurgery for fabrication of leaflets during surgical replacement of aortic valves. In some cases, leaflets are made of specially prepared decellularized bovine pericardium. However, of special interest is the use of patients’ own pericardium, in view of minimizing the rejection risk. For such operations, the excised pericardium is prepared according to the Ozaki protocol [66] comprising its chemical processing (cross-linking) in glutaraldehyde solution and simultaneous application of moderate tensile loading, which affects the tissue biomechanics. Conventionally, pericardium samples were studied in tensile tests [4,67,68], the result of which were not always consistent. Recently, in view of the positive experience with C-COE characterization of corneal samples having comparable with pericardium thickness, C-OCE was applied to compare human pericardium elasticity in several states [65]. 

It was shown in [65] that C-OCE made it possible to clearly distinguish elastic properties of human pericardium samples in four different states: (1) native-state samples; (2) chemically cross-linked without simultaneous stretching; (3) chemically processed with normal stretching; and (4) chemically processed samples subjected to excessive stretching, which could somewhat damage the tissue. Figure 5(a-1)–(a-4) show structural OCT images corresponding to those samples and Figure 5(b-1)–(b-4) present the corresponding stiffness maps reconstructed with C-OCE. The initially obtained stress–strain dependences are shown is Figure 5(c-1) by dashed lines, and the solid lines show the fitting curves. Despite clearly different forms of the stress–strain dependences, in all four cases they are well fitted using the nonlinear stress–strain dependence (19). The smallest slope (lowest Young’s modulus) is observed for pericardium in the native state. 

The other panels in Figure 5 are based on the analysis of the fitted stress–strain curves similarly shown in Figure 4. The derived values of the tangent Young’s moduli are plotted against compressive strain in Figure 5(c-2) and against the applied compressive stress in Figure 5(c-3). The reconstructed distributions υ(σ) and nonlinearity parameters β are shown in Figure 5(d-1),(d-2). It is clear from Figure 5(d-1) that the largest volume content of gaps/pores (the largest area under υ(σ) curve) occurs for the native state of pericardium. For the samples subjected to cross-linking in glutaraldehyde according to the Ozaki procedure, the content of compliant pores becomes strongly reduced, especially for the correctly prepared sample 3. This sample exhibits maximal stress sensitivity at fairly low stress level with the fastest trend of the Young’s modulus to the matrix value $E_{m}$. Correspondingly, the nonlinearity parameter β is largest for this sample with its maximum around 5 kPa stress. Then, it rapidly tends toward zero for stress > 10 kPa, where $E(\sigma )\to E_{m}$. Qualitatively similar behavior demonstrates chemically processed, but nonstretched sample 2 (in which the matrix value $E_{m}$ is ~4 times lower) and, in comparison with the correctly prepared sample, the aspect ratio of pores/gaps is larger because larger stresses are required to close the gaps. The overstretched sample 4 evidently experienced some damage (rupturing). In comparison with the correctly prepared sample 3, the overstretching evidently induced the appearance of additional breaks/gaps, which are clearly evident in the reconstructed broad distribution υ(σ) for this sample. Thus, it is not surprising that, except for the narrow region < 1 kPa, through most of the investigated stress range from ~1 kPa to 20 kPa, the overstretched sample 4 demonstrated a significantly lower Young’s modulus than the normally stretched sample. Qualitatively, pericardium and corneal samples demonstrate similar elastic behavior, which seems expectable bearing in mind that these tissue types are composed of stacks of collagenous layers.

### 3.3. Nonlinearity of Murine Tumor Samples

Certainly, along with such layered collagenous tissues as cornea and pericardium, for which the presence of interlayer gaps/pores was expected, it was interesting to apply the derived geophysics-inspired stress–strain law to other tissue types, for which microstructural features may significantly differ. In particular, from the very beginning of C-OCE development, much attention has been given to oncology-related problems, initially to the characterization of breast cancer samples [25,69,70]. It was not evident at all that the same fitting-stress dependences as for collagenous corneal and pericardium tissues would also give adequate results for tumors with a rather different microstructure. 

Here, we consider an example based on reprocessing data obtained in study [26] related to the C-OCE characterization of experimental murine tumors. Figure 6 shows the results of the C-OCE characterization of tumorous tissue in different states of tumors 4T1 inoculated at mice’s ears and subjected to chemotherapy (4T1 cell line is used as an animal model of mammary carcinoma). The meaning of panels (d-1)–(e-3) in Figure 6 is similar to those in Figure 4 for corneal tissue. The initially obtained stress–strain curves shown by dashed lines in Figure 6(d-1) again demonstrate that Equation (19) enables excellent fitting results shown by solid lines. In comparison with the viable tumor zones, the tumor regions with edema and especially necrotic tumor cells demonstrate strongly decreased stiffness. This is clear from the spatial stiffness maps in Figure 6(b-1)–(b-3), as well as from the slopes of the stress–strain curves shown in Figure 6(d-1) and the derived Figure 6(d-2),(d-3). 

It is interesting to note that similar to pores in the heated cornea in Figure 4(e-1), the reconstructed pore distribution υ(σ) in Figure 6(e-1) for necrotic tumor cells also has a maximum around 10 kPa. However, in Figure 6(e-1) for the necrotic tumor cells, the pores are not resolved, unlike that shown in the histological image for the heated cornea (see Figure 4c, lower image). In this regard, it should be pointed out that the properties of pores in terms of closing stress and strains are determined by their aspect ratios rather than the absolute sizes. Therefore, the mechanical response may be similar for either big or small pores if their distributions υ(σ) are similar. 

### 3.4. Nonlinearity of Patients’ Breast Cancer Samples

Breast tissues are very heterogeneous, some of them being composed mostly of collagen (like fibrous stroma), whereas other components of tumors are composed of cancer-cell agglomerates, in which collagenous fibers are almost absent, as directly confirmed by second-harmonic-generation optical microscopy [34]. Nevertheless, it was found that the stress–strain curves for breast cancer tissues also demonstrate surprisingly good results of fitting using the derived Equation (19). Detailed discussions on the usage of C-OCE-based nonlinear stress–strain dependences to differentiate various morphological components of breast cancer tissues can be found in ref. [27]. 

Although usually regions of tumor cells exhibit a higher Young’s modulus, more detailed studies indicated that the characteristic ranges of the tangent Young’s modulus values may exhibit significant overlap for different morphological components of breast tissues. In view of this, for more accurate diagnostics, one may perform differentiation on the plane of two parameters: Young’s modulus defined by Equation (20) and the nonlinearity parameter defined by Equation (21). It was shown in [27] that on such a plane it was possible to differentiate up to seven morphological components. At the same time the utilization of the differences in either Young’s modulus or the nonlinearity parameter taken separately was insufficient for discrimination of those components. Figure 7 shows histological images (panels 7(b-1)–7(b-6)) of these seven morphological components of human breast cancer tissues, whereas all other panels present results of their C-OCE-based characterization. In Figure 7 these morphological components are marked by numbers: 1—peritumoral adipose; 2—benign fibrous stroma; 3—invasive lobular carcinoma; 4—invasive ductal carcinova of scirrhous structure; 5—invasive ductal carcinoma (IDC) with solid-scirrhous structure; 6—zone of scattered individual tumor cells in dense fibrous stroma with hyalinosis of collagen fibers; and 7—stroma with pronounced hyalinosis. The zones, for which the stress–strain curves were obtained, are marked by rectangles in both stiffness maps (7(a-1)–7(a-6) and the corresponding histological images. Like in the previous examples in Figure 4, Figure 5 and Figure 6, structural OCT images of the discussed zones 1–7 did not demonstrate appreciable differences. In contrast, stiffness maps (plotted for 4 kPa stress) and presented in panels 7(a-1)–7(a-6) already demonstrate rather clear differences among these zones. In particular, zones of tumor cells can be very clearly distinguished in the C-OCE images from adipose and fibrous stroma. Detailed discussions related to medical aspects can be found in [27], so we do not reproduce them. 

For the present discussion, it is important to emphasize again that although the seven stress–strain curves in Figure 7(c-1) have visually very different forms, all of them are very well fitted by Equation (19). The other derived dependences are similar to those in Figure 4, Figure 5, Figure 6 and Figure 7 and show significant differences in terms of the reconstructed porosity distribution υ(σ), Young’s modulus E(σ), and nonlinearity parameter β. Despite the overlap in the individual parameters E(σ) and β for samples belonging to different tissue types, it was shown in [27] that on the (E,β) plane, it was possible to differentiate and automatically segment in C-OCE images all seven tissue types presented in Figure 7. In comparison to conventional time-consuming histology (requiring several days) and even significantly faster histology of frozen tissue, C-COE examination is carried out using freshly excised tissue samples and can be performed intraoperatively. In this regard, the possibility to obtain high-quality fitting for diverse a priori unknown tissue types using the fitting function of the same functional form is obviously important for the interpretation of elastographic data. 

### 3.5. Nonlinearity of Human Lymphatic Nodes

The next examples shown in Figure 8 are based on study [28] related to clarification of C-OCE possibilities to differentiate metastatic and nonmetastatic human lympatic nodes (LN) excised during breast cancer surgical operations. Without providing details of the medical aspects, Figure 8 presents results for the following LN types: 1—normal LN; 2—nonmetastatic LN with follicular hyperplasia; 3—nonmetastatic LN with sinus hytiocytosis; and 4—metastatic LN. In plots (d-1)–(d3) and (c-1)–(c-3) in Figure 8, the curves corresponding to these LN types are marked by numbers 1–4 and are shown in different colors.

Unlike the study reported in [28], where averaged curves for each group of LNs were shown, Figure 8 shows examples of such curves for individual LNs from each of the groups. Referring to [28] for a more detailed discussion of medical issues, here we point out that structural OCT scans shown in Figure 8(a-1)–(a-4) do not demonstrate strong differences, much like the structural scans for the other tissues. In contrast, the stiffness maps (plotted for 4 kPa stress) in Figure 8(a-1)–(a-4) exhibit rather clear differences that are similar to the previous examples of OCE maps provided in Figure 4, Figure 5, Figure 6, Figure 7 and Figure 8. 

The differences revealed in C-OCE-based stiffness maps are confirmed by histological images. Figure 8(d-1) shows the stress–strain curves for the discussed four types of LN (dashed lines). For all four LN types, Equation (19) again demonstrates very good fitting results (solid lines in Figure 8(d-1)). The other panels are derived from the fitted stress–strain curves similar to that shown in Figure 4, Figure 5, Figure 6 and Figure 7. Although the stiffness of LN in the normal state is clearly lower than for the other three LN types, for the latter LN-groups, an appreciable overlap of stiffness values was found. In view of this, for differentiation of reactive nonmetastatic and metastatic LNs, the Young’s modulus was used in combination with the nonlinearity parameter β [28]. For estimating the latter, the proposed stress–strain law (19) played the key role.

### 3.6. Elastic Nonlinearity of Small Intestine Tissues

In this section, we again present several examples related to nononcological tissues. These data were obtained in the course of C-OCE-based characterization of small intestine tissues in mini-pigs, which are often used in various biomedical studies because the structure of the intestine and many other organs is very similar to that in humans. The results of those studies were presented partially in [71], where C-OCE was used for the detection of emergence of tissue rupturing during procedures imitating intestine-wall distraction. It is known that the intestine wall structure is essentially heterogeneous, comprising the layers of muscular tissue, submucosa and mucosa. All of the layers may be in normal and pathological states and, to the best of our knowledge, the application of C-OCE for the first time made it possible to selectively characterize biomechanical properties of various layers in the studied intestine-wall layers. For the purpose of the present study, we focus exclusively on discussing the stress–strain dependences and the results of their fitting using the stress–strain law (19) derived in Section 2.1, rather than on the medical aspects of various states of the intestine. 

Representative results of the combined C-OCE and histological characterization of the intestine wall are presented in Figure 9. Column 1 shows the reconstructed C-OCE maps of the tangent Young’s modulus for the applied compressive stress 2 kPa. The corresponding histological images in approximately the same scale together with the names of the four tissue types 1–4 are shown in Figure 9 in column 2. Column 3 shows zoomed fragments of those histological images. Notice that the network of interconnected pores/gaps visible in these zoomed fragments is very similar to the images of interstitium discussed in ref. [61]. 

Figure 9(c-1) shows four stress–strain curves obtained using C-OCE for the muscular layer of the intestine wall (muscularis interna, label 1) and three different states of the inner mucous layer (labels 2–4). The dashed lines in panel 9(d-1) are experimental curves and the solid lines of different colors correspond to the fitting law given by Equation (19). All four stress–strain curves are pronouncedly nonlinear, including the curve for muscularis interna, for which even the zoomed histological image does not yet resolve visible pores/gaps, unlike the histological images of mucosa. Nevertheless, all four rather differently appearing curves in Figure 9(d-1) are very well fitted by Equation (19).

The other panels in Figure 9, similar to the previous examples, demonstrate the stiffness–strain (d-2) and stiffness–stress (d-3) curves; (d-4) shows the reconstructed pore distribution υ(σ) described by Equation (17) and (e-2) is the nonlinearity parameter β(σ) given by Equation (21). In Figure 9(e-3), the nonlinearity parameter β(σ) given by Equation (21) is plotted against the tangent Young’s modulus given by Equation (20). The main message from Figure 9 is that rather different forms of experimentally obtained stress–strain curves shown in panel (d-1) are very well fitted by the proposed stress–strain relation (19). It also worth noting that for all states of the mucosa layer, the histological images demonstrate existence of some interstitium very similar to that reported in [61].

### 3.7. Nonlinear Elastic Properties of Plaques

The last example of application of Equation (19) is also unrelated to oncology and presents the results of C-OCE characterization of various types of plaques developed inside patients’ blood vessels. The excised samples of such vessels were subjected to C-OCE characterization. The structure in Figure 10 is based on reprocessing of data from study [72] and is similar to Figure 8 and Figure 9. Namely, columns (a-1)–(a-5), (b-1)–(b-5) and (c-1)–(c-5) show structural OCT images, the derived Young’s modulus maps plotted for 4 kPa stress and corresponding histological images, respectively. Row (a-1), (b-1) and (c-1) relates to the normal vessel tissue. The other examples are for four types of plaques indicated in the caption of Figure 10. 

The initially obtained stress–strain dependences and fitting curves are shown in Figure 10(d-1) by the dashed and solid lines, respectively. The meaning of the other panels derived from Figure 10(d-1) is similar to those in Figure 4, Figure 5, Figure 6, Figure 7, Figure 8 and Figure 9. These examples again demonstrate that the stress–strain curves for various types of cholesterol plaques and normal vessel wall can also be well fitted using Equation (19), although visually the forms of the corresponding dependences are very different.

## 4. Discussion and Conclusions

We recall that initially the idea to utilize the analogy with description of elasticity of cracked rocks was stimulated by the results of C-OCE characterization of corneal tissue with clear layered structure. It was reasonable to attribute the observed pronounced stiffening at rather moderate compressive strains (~several percent) to stress-induced closing of some high-compliance structural features in the collagenous corneal tissue. In this strain range, the tangent Young’s modulus of cornea could strongly increase by a factor of 10–20 from a few tens of kPa up to ~one MPa. This strong modulus increase could be naturally explained by gradual closing of residual gaps (narrow pores) among collagen, which acted as high-compliance inclusions. The experiments performed on the modification of corneal microstructure with laser irradiation [57,62] made it possible to directly confirm the appearance of larger pores in histological images of irradiated corneas, for which the Young’s modulus decreased after laser heating. Furthermore, in [57], reasonably good agreement was obtained between the aspect ratios of pores visible in the histological images and estimations based on C-OCE data. 

The C-OCE examinations were always performed in the direction orthogonal to the orientation of corneal layers, so that the probing optical beam was aligned with the applied approximately uniaxial compression. The reaction of the crack-like gaps to this compression could be described in a simple 1D approximation. Analogous arguments also similarly appear reasonable for pericardium composed of collagenous layers. Thus, the similarity in the stress–strain curves for corneal tissue and pericardium and the possibility to closely fit those curves using the analogy with cracked rocks appeared quite reasonable. 

Next, we recall that for rocks, the stress dependence of elasticity is studied using both hydrostatic (“all-round”) compression, when all cracks are compressed normally to their planes independent of orientation, as well as applying uniaxial stress (e.g., using samples in the form of bars/rods and compressed axially). An important point is that while dry cracks are highly compliant with respect to either uniaxial or hydrostatic compression, in saturated materials the presence of water inside cracks drastically reduces their compressibility for hydrostatic compression. In contrast, for uniaxial compression, compliance of saturated cracks remains comparable with the dry case because the liquid can be squeezed out of the cracks. In this regard, in the examples considered above, biotissues certainly were saturated with water, but for the uniaxial compressive loading applied, even water-saturated structural features retained their high compliance. 

Another point worth mentioning is that in contrast to hydrostatic loading, for uniaxial compression, variously oriented crack-like defects experience different loading, comprising normal and tangential/shear components; furthermore, compliance of different types of actual cracks may comprise normal and shear compliance in different proportions [73]. Nevertheless, for given crack properties and moderately high crack density, the reduction in the different-type elastic moduli $M_{eff}/M_{m}$ can be written in a form qualitatively similar to Equations (11) or (13): $M_{eff}/M_{m}=1/(1+A_{m}n_{c})$. Here, $n_{c}$ is some effective concentration of cracks, and parameters $A_{m}$ differ somewhat for various types of the moduli but have the same order of magnitude [55]. Therefore, it was reasonable to expect that even for tissues that are unlike corneal tissue and not composed of a stack of parallel collagenous layers, the stress–strain curves functionally may also correspond to Equation (15) and probably even to Equation (19), if in the uncompressed state there are high-compliance features in the tissue structure. Indeed, the data presented in Section 3.4, Section 3.5, Section 3.6 and Section 3.7 confirmed that stress–strain curves for breast cancer tissues, lymphatic nodes, intestine tissues and plaques could be satisfactorily fitted by Equation (19), and thus qualitatively similar to the cases of collagenous cornea and pericardium.

Concerning Equation (19), we recall that it was derived from Equations (15) and (16) assuming the distribution υ(σ) of compliant inclusions over the closing pressure given by Equation (17). The form of that equation was also chosen by analogy with geophysics, where the exponential trend in Young’s modulus for compressed cracked rocks is often observed. Certainly, even if the elastic modulus is indeed reduced due to the presence of high-compliant inclusions, in principle, the distribution υ(σ) of the inclusions over closing pressures may differ from υ(σ) assumed in Equation (17). However, for other distributions of interstitial gaps/pores, one may also observe that the slope of the experimental stress–strain relationship σ(s) becomes steeper with increasing compression. In such a case, the stress dependence of $E_{eff}(\sigma )$ can also be obtained by differentiating the stress–strain relation, $E_{eff}(\sigma )=d\sigma /ds$. Equation (16) can then give the actual distribution υ(σ) of the interstitial gaps/pores that, in principle, may differ from Equation (17). 

Nevertheless, the examination of a broad range of strongly contrasting tissue types discussed in Section 4 demonstrates that Equation (19), based on the distribution υ(σ) described by Equation (17), gives very good results of fitting. The additional advantage is that the three fitting parameters in Equation (19) have a clear physical interpretation. Namely, parameter $E_{m}$ characterizes the Young’s modulus of the homogeneous matrix (host) tissue; dimensionless parameter $\upsilon_{t}$ corresponds to total specific volume content of the soft microstructural features; and finally, parameter B with the dimensionality of stress characterizes the stress sensitivity of the soft features. For the latter, the distribution υ(σ) over closing stresses is expected to obey Equation (17).

The proposed model is not totally phenomenological because is reflects the tissue microstructure, although the supposed soft features are described semi-phenomenologically, in terms of their compliance parameter ς, characteristic aspect ratio α and characteristic closing stress $\sigma_{clos}\sim \alpha E_{m}$ and strain $s_{clos}$ that are inter-related by Equations (6)–(8). In some cases, e.g., for corneal samples, in which the narrow crack-like pores were directly visualized, even the aspect ratios visible in the histological images reasonably agreed with the characteristic values inferred from the measured complementary modulus reduction and tissue dilatation caused by the laser-induced pores [57]. It can also be pointed out that our C-OCE observations of tissue elasticity response to compression, which indicates the existence of compliant pores in diverse tissues, agrees very well with the independent results reported in ref. [61]. In that study, microscopically examined frozen biopsy samples with well-preserved microstructure clearly exhibited the presence of numerous interstitial fluid-filled pores in rather diverse tissue types.

It worth noting that the compression applied to the tissue penetrated by liquid-filled pores is quasistatic, whereas the pores are not isolated from the surrounding tissue. Thus, during slow compression in C-OCE examinations the liquid rather freely squeezes from the pores into the surrounding tissue. Thus, the weak compressibility of the filling liquid does not appreciably affect the elastic response of the compressed pores, which remain highly compliant. A similar situation occurs in rocks when the sufficiently slowly compressed pores/cracks are interconnected and the saturating liquid is squeezed out of the crack-like pores and does not appreciably reduce their high compliance with respect to uniaxial compression.

For conditions of slow compression typical of C-OCE examinations, the results presented above demonstrate that the proposed approach is very well suited for fitting and interpretation of nonlinear stress–strain curves. It was also demonstrated for various tissues that the C-OCE method enables spatially resolved visualization of the Young’s modulus (as indicated in Figure 4, Figure 5, Figure 6, Figure 7, Figure 8, Figure 9 and Figure 10), allowing for automated segmentation of various morphological tissue components based on the differences in their elastic properties. These C-OCE capabilities are well corroborated by parallel histological examinations [25,26,27,34]. 

Further, using the ability of the C-OCE technique to obtain spatially resolved nonlinear stress–strain relations, one may simultaneously analyze the tangent Young’s modulus and nonlinearity parameter on the 2D plane to improve the accuracy of elasticity-based diagnostics. Similar prospects for tissue nonlinearity utilization also attract much attention in ultrasound elastography [29,30,31,32,33]. However, the C-OCE technique opens especially convenient possibilities in this regard and allows for improved spatially resolved differentiation of several morphological components or tissue types that are not reliably distinguished if their linear and nonlinear elastic parameters are analyzed separately [27,28]. 

We also emphasize that the examples of stress–strain curves and stiffness maps shown in Section 4 or discussed in studies [25,26,27,28] related to diagnostic applications of C-OCE; the examination of tissues was performed in the regime of quasistatic deformation, in which the influence of viscosity could be neglected. However, studying of tissue viscoelasticity also attracts significant attention in the literature, so that various rheological models are widely discussed. Qualitatively it is clear that, generally speaking, squeezing of saturating liquids from crack-like pores accounted for in the proposed model should have viscous character. Consequently, for higher rates of straining, viscoelasticity effects should give non-negligible contribution, so that the next possible direction in the development of the model presented above may be incorporation of viscoelasticity. In this regard, analogies with physics of fluid-saturated rocks may also be useful. 

It can also be noted that although narrow crack-like structural features can be viewed as a special case of porosity, it has been clearly understood in geophysics that mechanically the behavior of high-compliance pores with small aspect ratios is very different from the so-called “equant” pores (i.e., various channels with geometry close to cylinders or spheroidal voids). In view of this difference, geophysical models describing mechanics of rocks containing “soft” crack-like porosity were considered essentially independently from other poroelasticity models that mostly accounted for the presence of equant pores, which constitute “rigid” porosity. Since many biotissues are often penetrated by various channels (blood vessels and lymphatic vessels), importance of poroelasticity-based models in biomechanics certainly has been understood for years [35,42]. However, fairly “rigid” equant pores do not induce the pronounced small-strain nonlinearity under unified compression in contrast to demonstrations given in Section 4 for rather diverse tissues. Thus, by analogy with separate consideration of crack-containing rocks and other poroelastic models in geophysics, in biomechanics the “soft” and “rigid” pores should also be clearly distinguished.

Another important remark concerning alternative constitutive laws widely discussed in the pertinent literature is that they are mostly applied to interpretation of tensile tests of biotissues, like in [5], where even more complex combinations of straining types may be used, e.g., biaxial stretching. In such tests, at least sufficiently far from the damage threshold, biotissues often also exhibit stiffening. In contrast to the uniaxial compression in C-OCE, the role of various pores for tension-induced stiffening is less evident. However, it is reasonable to consider that properties of collagenous fiber/bundle networks are important for the modulus variation in tensile tests. In particular, such collagenous fibers/bundles usually exhibit pronounced wavy and even spiral structure, which is gradually unbended/straightened by increasing tensile loading. Mechanical modeling confirms that unbending/untangling of such initially curly/wavy structures under tension leads to pronounced stiffening of the material [35,41,74]. Similar remarks relate to the geometry of large biomolecules, which in the absence of loading also often appear as curled structures rather than straight chains of elastic elements. However, the tension-induced stiffening caused by straightening of such geometrical structures is very different from the tissue stiffening due to compression-induced closing of interstitial narrow pores considered in this study. In this regard, the constitutive equation proposed here for biotissues obtained by analogy with rock physics does not compete with earlier proposed models intended for the interpretation of biotissue response in tensile tests (like the influence of straightening of elastic fiber with initially wavy geometry [35,41,74]). In addition to the abovementioned difference between compression and tension, in many widely discussed biomechanical models, special attention was given to accounting of rather arbitrary types of tissue deformations. For example, in models based on Mooney–Rivlin or Ogden potentials and similar ones, the elastic energy is represented in terms of invariants of strain tensors. On the one hand, the invariant-based formulations make such models rather general in terms of accounted types of deformations. This universality is due to the fact that such models are essentially phenomenological and do not explicitly relate the nonlinearity of a material with its microstructure. In contrast, particular microstructural features usually demonstrate rather deformation-specific behavior. For example, the abovementioned presence of wavy fibers is obviously important for tissue nonlinearity under stretching, but cannot explain the pronounced small-strain nonlinearity under uniaxial compression demonstrated for many of the tissues described in Section 4. 

It is quite evident that models accounting for specific microstructural features cannot be as universal as phenomenological descriptions based on general symmetry properties, or requirements of invariance. In this regard, the proposed model represents a reasonable compromise since it is essentially based on accounting of high-compliance microstructural features in the matrix tissue, but at the same time these features are described without geometrical details that are too specific. Namely, they are characterized by compliance parameters, overall volume content and distribution over closing pressures, which already allows one to describe the pronounced nonlinearity of many rather different tissues subjected to small-strain uniaxial compression. This type of tissue loading is an important case, as it is often used in indentation tests and especially in the compression OCE technique developed in recent years.

Concerning the interpretation of C-OCE data, the proposed model Equations (15)–(19) have already proven to be highly efficient for high-quality fitting of experimental stress–strain dependences. This is very useful for the subsequent analysis of elastographic data to improve the accuracy of diagnostic conclusions. Evidently, such equations accounting for compression-induced closing of high-compliance structures may also be very useful for interpreting results obtained by nonoptical indentation methods, as described in [3], and similar techniques.

## Figures and Tables

**Figure 1 materials-17-05023-f001:**
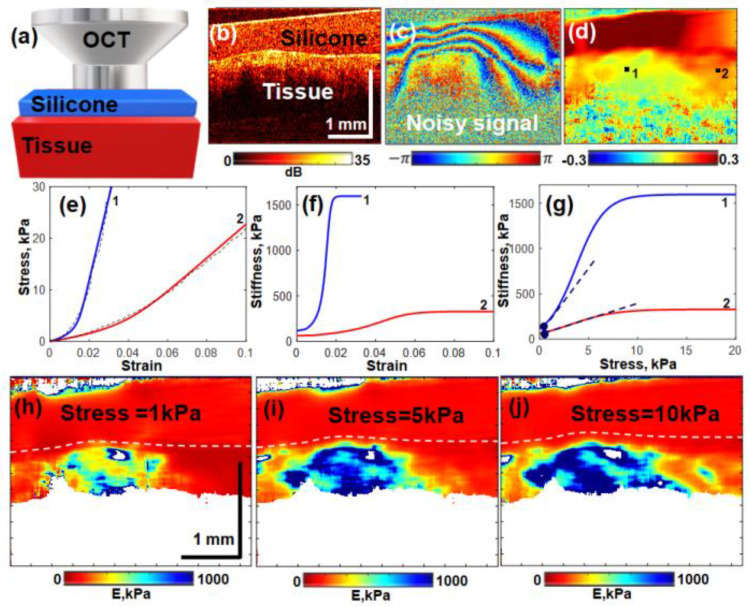
Principle of OCT-based strain visualization and utilization of reference layers of elastically linear silicone for obtaining stress–strain dependences of biological tissues: (**a**)—measurement configuration; (**b**)—structural OCT scan; (**c**)—color-coded interframe phase difference; (**d**)—reconstructed axial-strain map; (**e**)—stress–strain curves for regions marked 1 and 2 in panel (**d**) (dashed lines for experiment and solid ones for fitting); (**f**,**g**)—derived dependences of the Young’s modulus on strain and stress, respectively; (**h**–**j**)—distributions of the tangent Young’s modulus plotted for three different pre-chosen levels of applied stress to demonstrate significant modification of stiffness distribution caused by the tissue nonlinearity. The strong difference between the stress–strain curves obtained for a breast cancer sample in locations 1 and 2, shown in (**d**), is due to the fact that point 1 is close to the center of a stiff and strongly nonlinear agglomerate of cancer cells, whereas point 2 belongs to the peritumoral zone that has significantly lower stiffness and weaker elastic nonlinearity.

**Figure 2 materials-17-05023-f002:**
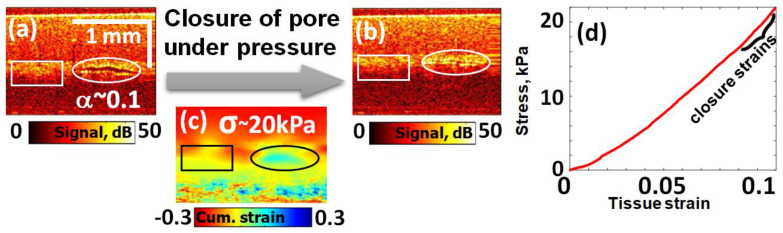
OCT visualization of closing of a narrow crack-like defect (with the position marked by ellipse) in a sample of pericardium. Panel (**a**) is the structural image in the initial open state of a crack-like defect with α~0.1; (**b**) is the image corresponding to the moment when the defect becomes visually closed and when the strain in the tissue estimated within the white rectangle attains $s_{clos}\sim \alpha \sim 0.1$; (**c**) is the map of cumulative strain up to the moment of the defect closure, demonstrating that in the vicinity of the crack interface the tissue remained uncompressed until the closure; (**d**) stress–strain dependence obtained aside the defect (region marked by the rectangle), the slope of which indicates that $E_{m}\sim 200$ kPa. The visual closure of the crack shown in (**b**) occurs for $\sigma_{clos}\sim 20$ kPa, in agreement with Equation (6) for α~0.1.

**Figure 3 materials-17-05023-f003:**
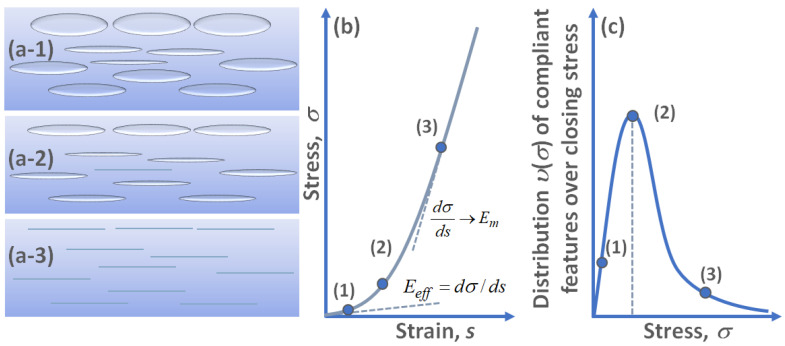
Schematic illustration of gradual closing of highly compliant crack-like defects with increasing stress (panels (**a-1**)–(**a-3**)): (**b**) is the complementary nonlinear stress–strain dependence σ(s) and (**c**) is bell-shape distribution υ(σ) of the defects over closing stress values. In state 1, almost all compliant defects are open and the effective modulus $E_{eff}=d\sigma /ds$ is minimal; in state 2, near υ(σ) maximum dependence σ(s) exhibits pronounced curvature; in state 3, almost all defects are closed, so that σ(s) tends toward a linear asymptotic behavior, whereas the modulus tends toward the matrix value $E_{m}$.

**Figure 4 materials-17-05023-f004:**
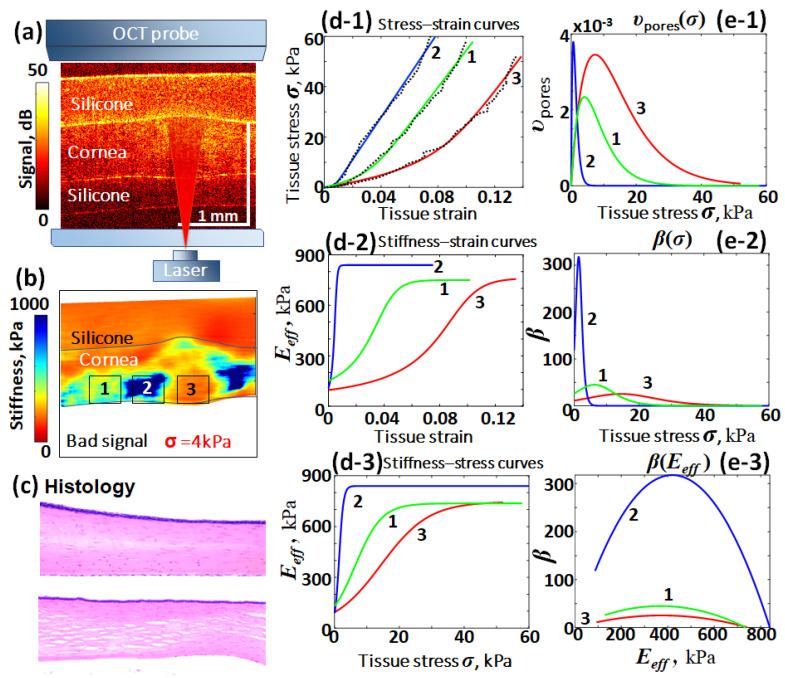
Example of C-COE-based and histological characterization of corneal tissue: (**a**) is the schematic of experimental configuration and a typical structural OCT scan of a corneal sample placed between two layers of reference silicone used for quantification of corneal stiffness; (**b**) shows the tangent Young’s modulus distribution (for 4 kPa stress) in silicone and cornea after heating with a laser beam passing through region 3; (**c**) are representative histological images of a nonheated corneal sample (upper image) and laser-heated one (lower image) in which a system of laser-induced narrow pores are visible; (**d-1**) shows the post-heating stress–strain curves selectively obtained in the center of heating zone 3, adjacent zone 2 and fairly distant nonheated zone 1 (dashed lines are experimental and solid lines are results of fitting using Equation (19)). Panels (**d-2**) and (**d-3**) show the tangent Young’s modulus versus strain and stress, respectively. Panel (**e-1**) shows the reconstructed pore distribution υ(σ) described by Equation (17); (**e-2**) is the nonlinearity parameter β(σ) given by Equation (21). In panel (**e-3**), the nonlinearity parameter β is plotted against the tangent Young’s modulus.

**Figure 5 materials-17-05023-f005:**
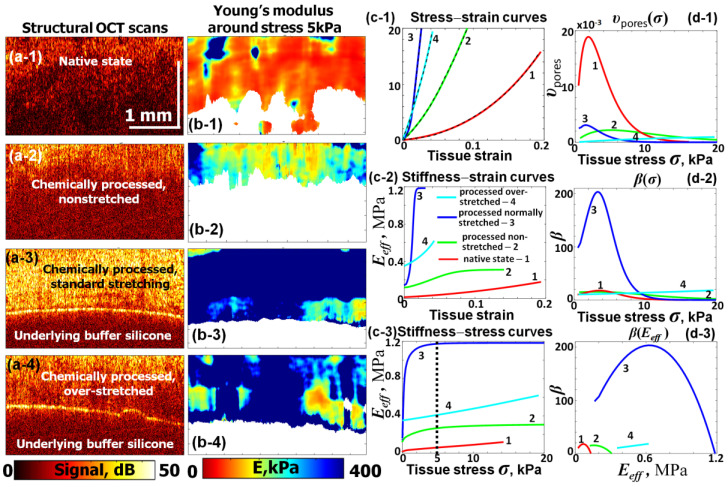
Results of C-OCE characterization of human pericardium in various states based on reprocessed data of [65]: ((**a-1**)–(**a-4**)) are structural OCT images; ((**b-1**)–(**b-4**)) show the spatial distribution of the Young’s modulus for 5 kPa applied stress; (**c-1**) shows the experimentally measured (dashed lines) stress–strain curves and solid lines are the results of fitting using Equation (19). The other plots (**c-2**),(**c-3**) and (**d-1**)–(**d-3**) are derived from the fitting curves in (**c-1**) like the similar plots in Figure 4. Decrease in thickness can be clearly seen for samples subjected to different tensile loading from nonstretched to overstretched during the chemical processing. For clarity, the upper reference silicone layer is removed from the images in panels (**a-1**)–(**a-4**) and (**b-1**)–(**b-4**).

**Figure 6 materials-17-05023-f006:**
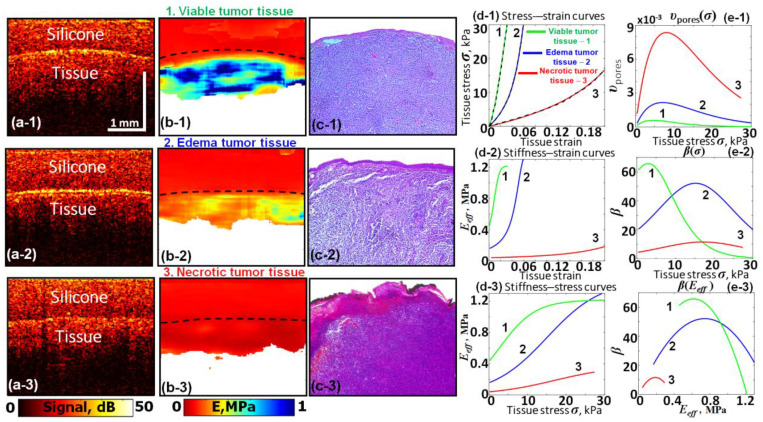
Results of C-OCE characterization of animal tumorous tissue in various states (based on reprocessed data from [26]: ((**a-1**)–(**a-3**)) are structural OCT images; ((**b-1**)–(**b-3**)) are the spatial distribution of the Young’s modulus for 4 kPa applied stress; ((**c-1**)–(**c-3**)) are the corresponding histological images; (**d-1**) shows the experimentally measured (dashed lines) stress–strain curves and solid lines are the results of fitting using Equation (19). The other plots (**d-2**),(**d-3**) and (**e-1**)–(**e-3**) are derived from the fitted curves in (**d-1**) like similar plots shown in Figure 4, Figure 5 and Figure 6. The highest stiffness is for viable tumor and the lowest for nectrotic tumor cells.

**Figure 7 materials-17-05023-f007:**
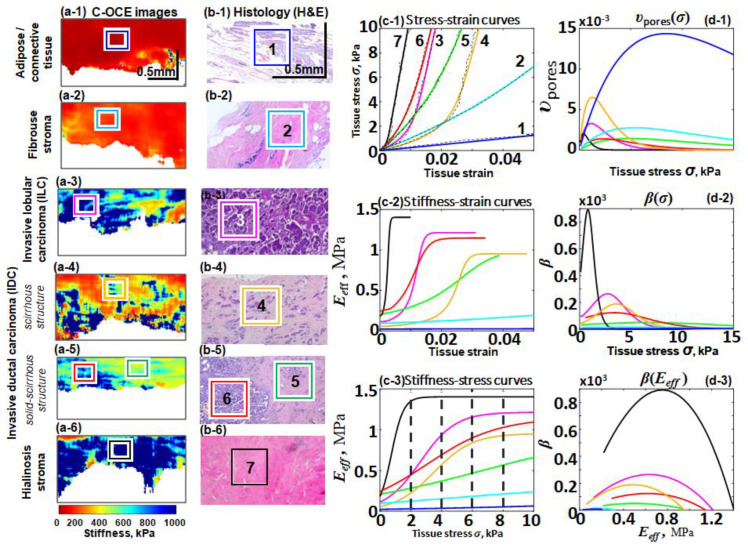
Results of C-OCE characterization of human breast cancer tissues (based on reprocessed data of [27]: ((**a-1**)–(**a-6**)) are the spatial distribution of the Young’s modulus for 4 kPa applied stress and ((**b-1**)–(**b-6**)) are the corresponding histological images, where rectangles indicate zones, for which seven stress–strain curves shown in (**c-1**) are obtained; in (**b-5**), the red rectangle indicates the fairly pure IDC tumor zone and the green rectangle indicates the zone with scattered individual tumor cells in dense fibrous stroma with hyalinosis of collagen fibers. The experimental stress–strain curves are shown in (**c-1**) by dashed lines and solid lines are the results of fitting using Equation (19). The other plots (**c-2**,**c-3**,**d-1**,**d-2**,**d-3**) are based on the analysis of fitted curves like the similar plots in Figure 4, Figure 5 and Figure 6. The highest stiffness is observed in hyalinosis zones and the lowest for adipose/connective tissue. Notice that the colors of the rectangles indicating zones 1–7 in the elastographic and histological images correspond to colors of the curves in plots (**c-1**)–(**c-3**) and (**d-1**)–(**d-3**). In panel (**c-1**) these curves are additionally numbered, but in the other panels the curves are marked only by colors, whereas numbering is omitted to avoid overloading of notations in the plots.

**Figure 8 materials-17-05023-f008:**
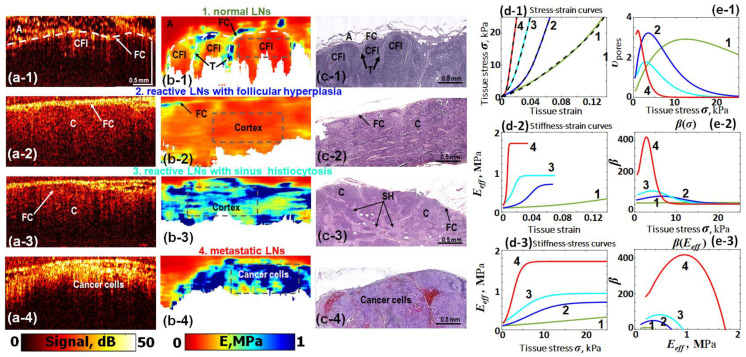
Results of C-OCE characterization of human LN excised during breast cancer-related surgeries (based on reprocessed data from [28]): ((**a-1**)–(**a-4**)) are structural OCT images for lymphatic nodes in normal state, LN with follicular hyperplasia (nonmetastatic), LN with sinus hytiocytosis (nonmetastatic), and metastatic LN; ((**b-1**)–(**b-4**)) are the corresponding stiffness maps for 4 kPa stress; ((**c-1**)–(**c-4**)) are the corresponding histological images, in which FC denotes fibrous capsule, CFl—cortical follicules, C—cortex of LN with follicular hyperplasia, SH—sinus histiocytosis and T—trabecules. The experimental stress–strain curves are shown in (**d-1**) by dashed lines, and solid lines are the results of fitting using Equation (19). The other plots (**d-2**),(**d-3**) and (**e-1**)–(**e-3**) are based on the analysis of fitted curves like the similar plots in Figure 4, Figure 5, Figure 6 and Figure 7. The highest stiffness is observed for metastatic LNs and the lowest for adipose/connective tissue.

**Figure 9 materials-17-05023-f009:**
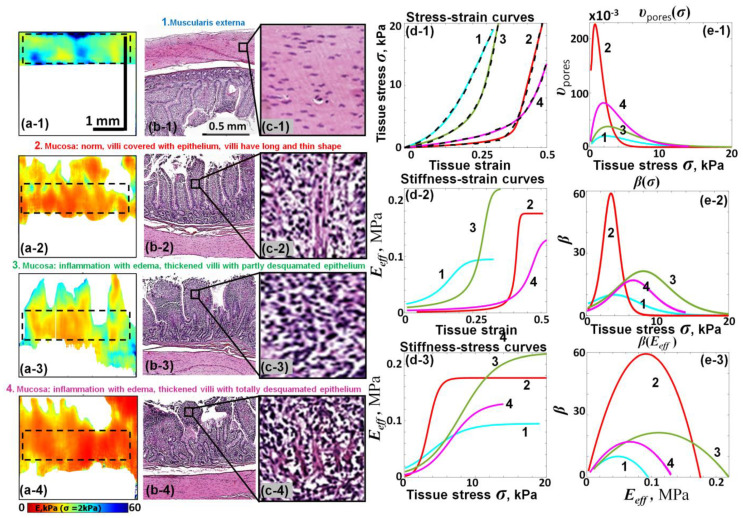
C-COE-based and histological characterization of small intestine wall comprising several layers in various states: ((**a-1**)–(**c-1**)) muscularis externa; ((**a-2**)–(**c-2**)) normal mucosa; ((**a-3**)–(**c-3**)) mucosa characterized by inflammation with edema, thickened villi with partially desquamated epithelium; ((**a-4**)–(**c-4**)) mucosa characterized by inflammation with edema, thickened villi with totally desquamated epithelium. Column (**a**) shows the corresponding maps of the tangent Young’s modulus for 2 kPa applied stress. Column (**b**) shows the corresponding histological images and column (**c**) presents zoomed fragments of images (**b**). Panel (**d-1**) shows the experimentally obtained stress–strain curves (dashed lines) and results of their fitting using Equation (19). The other plots (**d-2**), (**d-3**) and (**e-1**)–(**e-3**) are derived from the fitting curves shown in panel (**d-1**) like in Figure 4, Figure 5, Figure 6, Figure 7 and Figure 8.

**Figure 10 materials-17-05023-f010:**
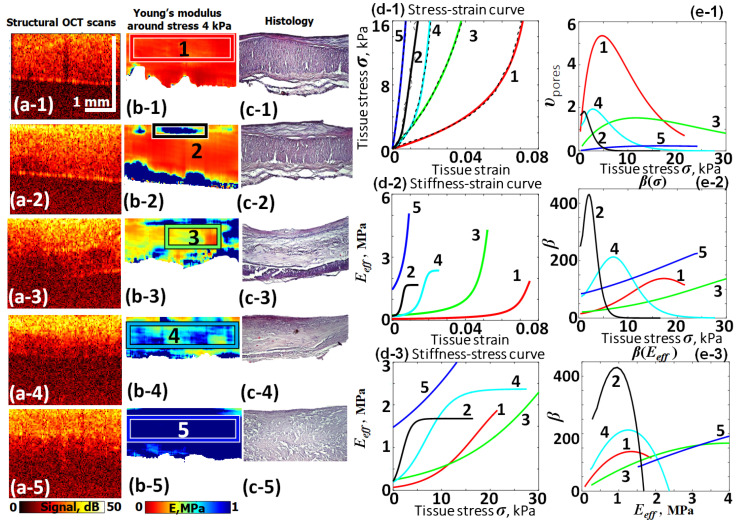
Results of C-OCE characterization of cholesterol plaques in excised fragments of human vessels (based on reprocessed data from [72]). Panels ((**a-1**)–(**a-5**)) show structural OCT images, ((**b-1**)–(**b-5**)) the corresponding stiffness maps (at 4 kPa stress) and ((**c-1**)–(**c-5**)) the corresponding histology: 1—for a vessel wall in normal state; 2—the wall with a lipid plaque; 3—sample with an unstable (and fairly soft at initial straining) cholesterol plaque; 4 and 5 are two examples with stiff cholesterol plaques with visible cholesterol crystals in histology, especially for 5. The experimental stress–strain curves for regions marked by rectangles in the stiffness maps are shown in (**d-1**) by dashed lines and solid lines are the results of fitting using Equation (19). The other plots (**d-2**)–(**d-3**) and (**e-1**)–(**e-3**) are based on the analysis of the fitted curves like the similar plots in Figure 4, Figure 5, Figure 6, Figure 7, Figure 8 and Figure 9. The sample numbers are indicated near the corresponding curves in panel (**d-1**)–(**d-3**) and (**e-1**)–(**e-3**).

## Data Availability

The raw data supporting the conclusions of this article will be made available by the authors on request.

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
