# Peer review of "Geophysics-Inspired Nonlinear Stress–Strain Law for Biological Tissues and Its Applications in Compression Optical Coherence Elastography"

_materials, 2024, doi:10.3390/ma17205023_

Round 1
Reviewer 1 Report
Comments and Suggestions for Authors
The authors present a constitutive model to describe the compressive behavior of biological tissues based on the evolution of pores-closing. The model is motivated by geophysics previous approaches and fitted to stress-strain curves taken from the literature, obtained by C-OCE techniques. The idea presented in original and new, although there are some aspects of the work that are not convincing/clear:
1. Previous modelling efforts based on poroelasticity introduce hyperelastic nonlinear formulations accounting for the water flow inside the tissue and reduction in apparent porosity with compression. These models describe the compressive behavior under quasi-static (and faster) loading conditions and, as they include hyperelastic energy potentials, also describe the nonlinear response under other loading conditions, e.g., tensile loading. The authors should comment on these models and clarify which are the benefits of their approach with respect to them.
2. It is not clear whether all experimental data is taken from literature or if some information is original from the authors. Some figure captions indicate the references from where the data is taken, but many others do not indicate anything. The authors should clarify this much more clearly in the manuscript.
3. If this is the case that all experimental data and postprocessing is directly taken from the literature, section 2.1 should be significantly reduced as this is state-of-art information that is already available in the literature and not a contribution from this work. The essential information can be kept saving a lot of space that, in the present form, distracts a lot from the real contribution of the article.
4. Section 2 jumps from 2.1 to 2.3, not sure if this is a typo or the authors missed section 2.2.
5. The stress-strain result in Figure 2d shows a quite linear response of the tissue. I think that if the pore closing would have that relevant impact on the compressive response of the tissue, a change in slope (stiffness) would be appreciated in the curve within the closure strain region. The authors may comment on this.
6. I do not really understand the shape of volume content evolution with stress. Why does it increase with stress in the first region? Please discuss this further. If the authors can provide evidence of these dynamics the work would be significantly strengthen, as the model would provide estimations of porous microstructure evolution during tissue deformation without the need of more expensive approaches.
7. The authors should present the fitting parameters used in the cases shown in section 3. Regarding this, which is the fitting connection with the microstructure? In case this is purely phenomenological, which is the advantage of this approach with respect to previous hyperelastic-based models in the literature? One major issue I find is that the current model assumes the matrix stiffness as a constant (E_m) and this has been demonstrated to be nonlinear (bulk material). For reference, the authors may look into previous work by, e.g., Silvia Buddays’ group on brain tissue.
8. The authors relate all the modelling formulation to changes in porosity, however, there are other microstructural features that highly impact the apparent stiffness of the tissue. For example: i) deformation and buckling of collagen fibers; ii) cytoskeleton remodeling of the cells within the tissue. For the former, there are several works in the literature. For the latter, the authors may look into recent mechanobiology studies (e.g., https://doi.org/10.1002/adma.202312497). These aspects should be discussed.
Comments on the Quality of English LanguageThe writing of the manuscript should be improved. There are many parts that can be significantly reduced (e.g., introduction or section 2.1), and many paragraphs (e.g., introduction) can be unified into single paragraphs to enhance the reading.
In addition, there are many typos.
Reviewer 2 Report
Comments and Suggestions for Authors
The article considers a generalization of a nonlinear elastic constitutive law taken from the geophysics context. The model predictions are compared with experimental data obtained through Quasistatic Compression Optical Coherence Elastography. The article is interesting, well-written, and aside from some required nuances, it presents no problems and has merit for publication.
INTRODUCTION
The introduction discusses AFM, which is largely irrelevant for biological tissues, and provides two references; however, the biological models only have one reference (there is no mention of inelasticity, microfiber rupture, viscoelasticity, or damage models. The authors should mention some of these models and compare them with their own, providing additional references). The summary of elastography methods is quite complete and appropriate. From the mention of collagen fibrils in section 2.3, it can be inferred that the study is conducted on soft tissue, but it should be clearly stated earlier which tissue is used and from which animal species, so that the results can be compared with the scientific literature.
DATA AND METHODS
In the text between equations (1) and (2), the relationship s = dU/dz is mentioned, which is more than adequate for rocks or bones due to small deformations. However, in soft tissues, the Green-Lagrange strain tensor would be reduced to s = dU/dz + ½(dU/dz)^2, so for nonlinear laws, the initial relationship [s = dU/dz] could contain significant errors for strains greater than 20%, which are not uncommon in soft tissues. Figure 1 considers regions 1 and 2, which show very different stiffnesses, but it is not clarified or explained why two adjacent regions of the same tissue present such significant differences in stiffness.
Regarding the stress calculation, it seems that the assumption is that the stress in the silicone is exactly equal to the stress on the surface of the tissue in contact with the silicone. However, at a certain depth, due to tissue inhomogeneities, there could be a significant difference between the internal stress in the biological tissue and the stress in the silicone at the surface. This should be clarified, explained, and potentially an error bound for this approximation should be presented.
In line 311, gaps/pores are mentioned in the collagenous tissue, which does not seem to correspond to reality. Typically, the spaces between collagen fibers are filled with an elastin matrix, whose structure depends on the tissue. Perhaps what the authors consider gaps are actually a much more deformable elastin matrix; therefore, the text referring to gaps or pores should be rewritten more realistically. Figure 2 seems to show such a pore, but since there is no information about the tissue or where the image comes from, it is doubtful whether this is the necessary type of evidence. The existence of defects similar to cracks in rocks within biological tissues is questionable (or should be clarified) and should be supported by substantial real evidence; it would be insufficient to say that the constitutive equation, deduced under the assumption that something like cracks exists in collagenous tissue, seems phenomenologically adequate.
Comparing equation (9) with (11), it appears that a factor 1/(zmax - zmin) would be necessary before the integral to yield some sort of average. The authors use a long paragraph (lines 402-433) to justify the validity of certain previous assumptions, but the evidence mentioned is a specific study, and the summary provided is not convincing. Thus, the claim in line 434 that "after demonstration of the utility of the idea" should be nuanced. In any case, the previous evidence suggests that the idea is useful, but it is not a complete demonstration in itself.
Equation (19) is nonlinear for moderate stress values, and for high values, it has an inclined asymptote. While it may be adequate for compression, this equation is entirely unsuitable for tension or other types of complex loading (this should be mentioned in the limitations).
RESULTS
The results for rabbit cornea, pericardial tissue, breast soft tissue, and human lymphatic nodes are interesting and show that the stress-strain response is as expected. Equations (19) and (20) provide a reasonable model of the stress-strain curves under compression of these tissues.
CONCLUSION AND DISCUSSION
Line 974 discusses tension in biological tissues. It is not explicitly stated here, but equation (19) would not reproduce the observed behavior (absence of an inclined asymptote). Additionally, the claim in line 976, "In such tests, at least sufficiently far from the damage threshold, biotissues usually also exhibit stiffening," is true for soft biotissues but clearly false for bones subjected to tension or bending, for example, where the stress-strain curve is concave. It is suggested to clarify that certain statements are only valid for soft biological tissues. Finally, the paper does not address how to generalize the proposed constitutive law. A proper constitutive equation or model explains mechanical behavior under any type of loading, whereas the proposed model is valid under uniaxial compression. A comment should be included to indicate the scope of the proposed law.
Reviewer 3 Report
Comments and Suggestions for Authors
The authors propose a nonlinear stress-strain law to describe the elastic properties of biological tissues. The proposed law is obtained through an analogy with the nonlinear constitutive laws for rocks with cracks/gaps, an approach supported by the presence of interstitial gaps/pores in biological tissues, as confirmed by histology. After presenting the derivation of the proposed nonlinear law, the authors demonstrate that the proposed approach is very well suited for fitting of nonlinear stress-strain curves obtained by quasistatic Compression Optical Coherence Elastography (C-OCE), by presenting an impressive array of studies on samples of various biological tissues, with excellent fits. The methods developed by the authors also has the advantage of each of the fitting parameters having a clear physical meaning.
This is a very innovative, highly relevant paper that clearly deserves to be published. The methodology is robust and executed with care and rigor. The conclusions reached by the authors are well supported by the authors.
In my opinion, the manuscript just needs an overall check concerning its writing. There are many typos and small errors, and some paragraphs should be rewritten to increase their readability. Apart from this, I only a few suggestions that, in my opinion, might improve the paper.
1. Please indicate that equation 2 assumes an optically homogenous medium (n constant).
2. By direct observation of the stress-strain curves for the several presented examples concerning biological tissues, it seems that the fit of those curves is less successful for the rabbit’s corneal samples, particularly for higher values of applied stress. Do you confirm this observation? If so, why this occurs? Is it a consequence of a wider range of stress? Do you expect a stress limit for the applicability of the proposed nonlinear stress-strain law?
3. Also, I believe it would be useful to provide an indicator of the goodness of the presented fits (reduced chi-square or other).
4. In my opinion, it would be informative to add a scale to the OCT images of Figures 2, 6 and 10 and to the OCE images of Figure 9.
5. In the text between lines 373 and 381, it is not clear to me if the symbol “υ” (upsilon) stands for the total volume of crack or to the ratio of the total volume of the cracks to the total volume of the tissue. To me, using the word “content” in line 374 and the word “specific” in line 381 generates confusion and degrades the clarity of the text. Moreover, later it is used the symbol υt for the total volume content of all cracks (line 392). I believe that the clarity of this section can be improved by improving the coherence of the terms and symbols used.
6. In line 397, I believe it should be “Eq. (11) can be rewritten” and not Eq. (12).
7. The paragraph comprising lines 545 to 551 is very confusing due to weak English usage. It seems that something is missing from the first sentence of the paragraph. Also, something is missing in the sentence in lines 563-564. Please verify if that is the case.
8. Please indicate what is a tumor 4T-1. Namely indicate that it is an animal model for stage IV human breast cancer.
Comments on the Quality of English Language
As I said, the manuscript will benefit from an overall check concerning its writing. There are some paragraphs should be rewritten to increase their readability. See above my comments 5 and 7. Here I also list some typos and small errors that I found:
Line 63: There are two typo errors: “shera-wave” and “visualized”.
Line 95: Typo error: “is compression USE”.
Line 174: Error: “the efficiently”.
Line 215: Typo: “thios”.
Line 227: Typo: “cuves”.
Line 333: Different style of reference (Mavko&Nur 1978).
Line 373: Typo: “incusions”.
Line 400: Typo: ”compilance”.
Line 824: Repeated words: “structure of intestine intestine”.
Line 943: Typo: “in principles”.
Line 952: Typo: “in rathe”.
Line 961: Mistake: “nonlinear stress-stress curves”.
Reviewer 4 Report
Comments and Suggestions for Authors
In this study, the authors developed a nonlinear stress-strain law inspired by geophysics to describe the elastic properties of biological tissues. By drawing analogies between crack closure in rocks and interstitial gaps in tissues, they proposed a constitutive model that fits experimental stress-strain data obtained through compression optical coherence elastography (C-OCE). The model's fitting parameters were shown to have clear physical meaning and demonstrated high diagnostic value, particularly in differentiating between various types of cancerous and non-cancerous tissues. The study presents a novel approach to understanding tissue mechanics through this geophysical analogy.
1) Authors are encouraged to include more experimental results or simulations that directly compare the stress-strain behavior of biological tissues and rocks under similar loading conditions. Alternatively, they can consider providing a more detailed justification for how biological microstructures resemble the cracks in rocks beyond general pore comparisons.
2) While the proposed nonlinear model fits the experimental stress-strain curves, the manuscript somewhat lacks robust validation against existing biomechanical models that describe soft tissues. Comparisons with other established models (e.g., hyperelastic models like the Ogden model or the Fung model) are not discussed sufficiently. Hence the authors are advised to include a quantitative comparison of the proposed model with existing biomechanical models, particularly those commonly used in soft tissue mechanics. This will help validate the proposed approach and demonstrate its advantages or limitations.
3) Authors have focused well on C-OCE, while this alternative can be powerful, the current version of the manuscript does not discuss sufficiently whether other methods like indentation or tensile testing could corroborate the findings. Authors could expand the discussion to include comparisons with other experimental techniques used in tissue biomechanics, such as tensile tests, ultrasound electrography, or AFM. Also, they can elaborate more in suggesting future directions for new experiments that could be performed using these techniques to support the nonlinear stress-strain law in a broader context.
4) Authors have explored various biological tissues like cornea and pericardium, but the current version somewhat lacks in significant variability in mechanical properties between different tissue types and how this affects the applicability of the model. By including a more detailed discussion on the range of tissue types and their mechanical properties can address this. Also, if the model is universally applicable or only suited for certain tissue categories (e.g., highly collagenous tissues). Adding more examples from diverse tissue types could strengthen the validity of the model.
5) Authors have included histological references to support the analogy between tissue porosity and cracked rocks, but it is to be noted that the histological evidence cannot completely be integrated with the stress-strain analysis. Hence, can the authors Provide clearer and more systematic links between histological images and the mechanical data? For each tissue type discussed, show how specific histological features (e.g., collagen fiber arrangement, pore sizes) correlate with the proposed stress-strain law. This will provide more concrete evidence for the model.
Round 2
Reviewer 4 Report
Comments and Suggestions for Authors
The authors have addressed all comments raised, provided sufficient explanations to questions raised, and made necessary changes to the revised manuscript where needed.
As a general suggestion for future works, it is recommended that authors include their in-text inclusions in their cover letters along with page numbers and line numbers in their cover letter to reviewers.